# Synergetic Aerosol Layer Observation After the 2015 Calbuco Volcanic Eruption Event

**Fábio J. S. Lopes** [1,*] , **Jonatan João Silva** [1,2], **Juan Carlos Antuña Marrero** [3], **Ghassan Taha** [4] and **Eduardo Landulfo** [1]

[1] Center for Lasers and Applications (CLA), Nuclear and Energy Research Institute (IPEN), São Paulo (SP) 05508-000, Brazil; silva.jonatan@ufob.edu.br (J.J.S.); elandulf@ipen.br (E.L.)

[2] Center for Exact Sciences and Technologies, Federal University of Western Bahia (UFOB), Barreiras (BA) 47801-275, Brazil

[3] The Atmospheric Optics Group of Camagüey (GOAC), Camagüey Meteorologic Center, INSMET, Camagüey, Cuba; anadelia@caonao.cu

[4] Universities Space Research Association, Greenbelt, MD 20771, USA; ghassan.taha-1@nasa.gov

[*] Correspondence: fabiolopes@usp.br; Tel.: +55-11-3133-9361

**Abstract:** On 22 April 2015, the Calbuco volcano in Chile (Lat: 41.33°S, Long: 72.62°W) erupted after 43 years of inactivity followed by a great amount of aerosol injection into the atmosphere. The pyroclastic material dispersed into the atmosphere posed a potential threat to aviation traffic and air quality over affected a large area. The plumes and debris spread from its location to Patagonian and Pampean regions, reaching the Atlantic and Pacific Oceans and neighboring countries, such as Argentina, Brazil and Uruguay, driven by the westerly winds at these latitudes. The presence of volcanic aerosol layers could be identified promptly at the proximities of Calbuco and afterwards by remote sensing using satellites and lidars in the path of the dispersed aerosols. The Cloud-Aerosol Lidar and Pathfinder Satellite Observations (CALIPSO), Moderate Resolution Imaging Spectroradiometer (MODIS) on board of AQUA/TERRA satellites and Ozone Mapping and Profiler Suite (OMPS) on board of Suomi National Polar-orbiting Partnership (Suomi NPP) satellite were the space platforms used to track the injected layers and a multi-channel lidar system from Latin America Lidar Network (LALINET) SPU Lidar station in South America allowed us to get the spatial and temporal distribution of Calbuco ashes after its occurrence. The SPU lidar stations co-located Aerosol Robotic Network (AERONET) sunphotometers to help in the optical characterization. Here, we present the volcanic layer transported over São Paulo area and the detection of aerosol plume between 18 and 20 km. The path traveled by the volcanic aerosol to reach the Metropolitan Area of São Paulo (MASP) was tracked by CALIPSO and the aerosol optical and geometrical properties were retrieved at some points to monitor the plume evolution. Total attenuated backscatter profile at 532 nm obtained by CALIPSO revealed the height range extension of the aerosol plume between 18 and 20 km and are in agreement with SPU lidar range corrected signal at 532 nm. The daily evolution of Aerosol Optical Depth (AOD) at 532 and 355 nm, retrieved from AERONET sunphotometer, showed a substantial increasing on 27 April, the day of the volcanic plume detection at Metropolitan Area of São Paulo (MASP), achieving values of $0.33 \pm 0.16$ and $0.22 \pm 0.09$ at 355 and 532 nm, respectively. AERONET aerosol size distribution was dominated by fine mode aerosol over coarse mode, especially on 27 and 28 April. The space and time coincident aerosol extinction profiles from SPU lidar station and OMPS LP from the Calbuco eruption conducted on 27 April agreed on the double layer structure. The main objective of this study was the application of the transmittance method, using the Platt formalism, to calculate the optical and physical properties of volcanic plume, i.e., aerosol bottom and top altitude, the aerosol optical depth and lidar ratio. The aerosol plume was detected between 18 and 19.3 km, with AOD value of 0.159 at 532 nm and Ånsgtröm exponent of $0.61 \pm 0.58$. The lidar ratio retrieved was $76 \pm 27$ sr and $63 \pm 21$ sr at 532 and 355 nm, respectively.

Considering the values of these parameters, the Calbuco volcanic aerosol layers could be classified as sulfates with some ash type.

**Keywords:** volcanic ash; Calbuco; Lidar; CALIPSO; MODIS; AERONET; LALINET

---

## 1. Introduction

In climate models, dealing with volcanic matter injected into the atmosphere is still a complex problem due to temporal and spatial variability, mass and volume of the gases ($CO_2$, water vapor, $SO_2$, and halogen compounds), accountability and due to the lack of a complete dataset capable of providing enough information of statistical robustness for future simulations. The injection point into the atmosphere, i.e., tropospheric or stratospheric, is also an important aspect of its influence on many distinct cycles such as hydrological, carbon and biogeochemical cycles [1]. As a weather and climate response driver, the effects are many such as diurnal cycle reduction, tropical precipitation decrease, summer cooling of NH tropics and subtropics, stratospheric warming, winter warming of Northern Hemisphere (NH) land, global cooling and ozone–UV depletion–enhancement [2,3]. In terms of radiative forcing (RF), values span from $-0.15$ to $-0.06$ W·m$^{-2}$ over the last 30 years, except for the Mt Pinatubo eruption in 1991, which had a RF of $-2.3 \pm 1.0$ W·m$^{-2}$ [4].

The Volcanic Explosivity Index (VEI) is a scale to estimate the explosive character of eruptions, and it is based on several parameters: the volume of ejecta in m$^3$; the intensity of eruption represented by the volume of ejecta per unit of time; the dispersive power related to the altitude of the column height in km; violence, which is related to the release rate of kinetic energy; and the destructive potential that depends of the extent of devastation [5]. According to these parameters, an eruption can be assigned a VEl value of 0 to 8.

Over the past eight years, there were many small eruptions with small impact on RF, and the majority did not inject a considerable amount of tephra and gases into the stratosphere. However, since 2010, only one eruption with Volcanic Explosivity Index (VEI) 5 and nine eruptions with VEI 4 occurred around the world; three of them occurred on the South America continent: Puyehue-Cordon Caulle in June 2011 (VEI 5) and Calbuco in April 2015 (VEI 4) in Chile, and Wolf in May 2015 in Ecuador (VEI 4). It is also important to mention that other intense volcanic explosions occurred around the globe, such as Kelut volcano in February 2014 in Indonesia [6,7], Nabro volcano in June 2011 in Eritrea [8,9] and Eyjafjallajökull in March 2010 in Iceland [10]. In general, those eruptions had some degree of large scale direct and indirect impacts as the pyroclastic material dispersed into the atmosphere poses a threat to aviation traffic, air quality deterioration and climatic radiative effects. In this paper, we discuss the remote sensing ability in South America (SA) to detect volcanic eruption material right after two large eruption events, one outside the continent and the other within SA boundaries.

The use of satellites is a robust way to build a database for stratospheric and upper tropospheric aerosols, which are mostly from volcanic origin. These databases are built with data from long and short term missions, such as OSIRIS, CALIPSO and SAGE II [11], as well as the recent ESA mission ADM-Aeolus [12,13]. In general, most of these databases are related to Pinatubo period but also cover other eruption events. The retrievals by such missions and platforms aim to optically characterize these aerosols giving results on backscatter and extinction coefficients, particulate depolarization and lidar ratios [14,15]. In some instances, the plume height information is a substantial information to have an idea of the upper troposphere/lower stratosphere mass exchange rate [16]. Other instrument networks such sunphotometers and Brewer spectrophotometers also give valuable information on volcanic plumes and supplement satellite missions on creating a consistent atmospheric weighting functions [17].

On 22 April 2015, after 43 years of inactivity, the Calbuco volcano in Chile (Lat: 41.33°S, Long: 72.62°W) injected a large amount of volcanic ash aerosols into the atmosphere from a VEI 5 eruption.

The pyroclastic material dispersed into the atmosphere posed at first a threat to aviation traffic and air quality, which prompted an alert in a large area, from the volcano location to Patagonian and Pampean regions, in Argentina, Chile and Paraguay. Due the general air masses circulation, the volcanic aerosol plumes traveled northeastward reaching the neighboring countries, Uruguay and Brazil. After several days, the volcanic ash from Calbuco crossed the Atlantic Ocean, reached the southern region of the African continent and was detected on Reunion Island between 18 and 21 km [18]. The presence of volcanic aerosol layers could be identified promptly at the proximities of Calbuco as well as by sensing techniques, both from space and ground, with measurements obtained in the path of the dispersed aerosols. CALIPSO, MODIS and OMPS onboard of Suomi NPP satellite were the space platforms used to track the injected layers and lidar systems from LALINET network in South America allowed us to get 4-D distribution of Calbuco ashes after its occurrence (22–30 April). Most lidar stations have collocated AERONET sunphotometers to help in the optical characterization but not all LALINET stations were able to observe this event given the air circulation pattern dominating this part of the globe and their distance from the atmospheric injection. Here, we present the volcanic layer transported over São Paulo area where a station is located showing the presence of erupted material around 19 km. The detection of volcanic plume in April 2015 at SPU LALINET station was the first event of this kind for this station and with the aid of remote sensing by means of remote sensing platform data, namely CALIPSO, MODIS, OMPS LP, the AERONET sunphotometer and SPU lidar system, we were able to estimate the sole plume AOD, its optical properties and retrieve the plume transmission, AOD, extinction and backscatter profiles, and Lidar ratio values to classify the volcanic aerosol. In addition, this observation will help to understand the mass transfer between the troposphere and lower stratosphere and models related to that [19]. This observation should be an initial effort to create a lidar-based observation database similar to those available in many sites around the globe [20–22].

In this paper, we present the tracking of the volcanic plume right after its release into the atmosphere up to about five days later when it was detected over the SPU lidar station in São Paulo. As shown below, the plume at some point bisected and one portion travelled eastwards towards the South African region and the Indian Ocean [18,23], while the other travelled to the northeast and reached the São Paulo region. This observation is important to distinguish between the two portions and, despite the good amount of reports to the one travelling eastwards, the detection of the volcanic plume proved to be of equal interest in its characterization. It is also worth mentioning that residence time of the released material, about 0.4 Tg, had a climate drive on the ozone hole in the southern polar region [24].

## 2. Methods and Instruments

The instruments on board each satellite platform are presented in the next sections and ground-based instruments such as lidar and collocated sunphotometer are discussed as well. The synergetic use of these platforms helped to understand the impact of the volcanic plume in terms of its optical properties.

### 2.1. LALINET SPU-Station

A Raman lidar system installed at the Nuclear and Energy Research Institute (IPEN), São Paulo (Brazil), was employed to measure profiles of the particle extinction and backscatter coefficients. Lidars are instruments that use a laser beam as source [25]. For the lidar system at SPU station, a commercial Nd: YAG laser operating at 1064, 532 and 355 nm is used. The laser energy per pulse is about 600, 400 and 230 mJ at 1064, 532 and 355 nm, respectively, at a repetition rate of 10 Hz and pulse durations of approximately 5 ns. The exit beam is expanded by a factor of 5 to achieve a divergence of less than 0.5 mrad. The outgoing laser beam is vertically directed to the atmosphere and the backscattered radiation is collected with a coaxial newtonian telescope that has a primary mirror diameter of 300 mm and a focal length of 1.5 m. The receiver field of view is set

to 0.1 mrad, providing a complete overlap between the laser beam and the telescope field of view at altitudes close to 300 m above the lidar system. The photons elastically backscattered at the 355, 532 and 1064 nm wavelength and the photons inelastically (Raman) scattered by nitrogen molecules are detected at 387 and 530 nm, and by water vapor molecules at 408 nm using photomultiplier tubes (PMTs, Hamamatsu type R9880U-110). All these backscattered signals are filtered and the background suppression is achieved by means of interference filters with a FWHM of 1 nm at the elastic channels and 0.25 nm for the inelastic ones. A commercial transient recorder operates in analog and photon counting modes recording data in 12-bit resolution. Data are averaged every 1 min, with a typical height resolution of 7.5 m. This system belongs to LALINET (http://lalinet.org), a federative coordinated lidar network focused on the vertically-resolved monitoring of the particle optical properties distribution over Latin America. The lidar data applied to this study were obtained during the most of 27 April, from 12:22 to 21:10 (UTC). The raw lidar profile was pre-processed and averaged for the whole period from when the SPU lidar station detected a thin layer at the lower stratosphere from a very noisy signal profile. Each different lidar technique has a different impact on the retrieval [26], and, to derive the physical and optical products from this thin volcanic aerosol, the transmittance method [27] was applied to determine the aerosol layer optical depth and lidar ratio. We also applied the Klett–Fernald–Sasano [28–30] inversion method by tuning the initial lidar ratio assumption with the AOD values retrieved from AERONET to obtain the particle backscatter profiles at 532 and 355 nm. From this profile, we calculated the integrated backscatter value of the aerosol plume to apply to the transmittance methodology.

## 2.2. AERONET Sunphotometer

The AErosol RObotic NETwork (AERONET) [31] is the NASA sunphotometer global network that, with more than 1000 instruments deployed around the globe, provides automatic sun and sky scanning measurements. Using direct sun measurements, AERONET provides both AOD and the Ångström Exponent (Å), which gives the wavelength dependence of the AOD. By using multiangular and multispectral measurements of atmospheric radiance and applying a flexible inversion algorithm [32], the AERONET data can also provide several additional aerosol optical and microphysical parameters, such as size distributions, single-scattering albedo and refractive index. The operating principle of this system is to acquire aureole and sky radiance observations using a large number of solar scattering angles through a constant aerosol profile, and thus retrieve the aerosol size distribution, the phase function and the AOD [33]. AERONET sunphotometer data from São Paulo station were retrieved to derive the AOD values at 532 and 355 nm and check how the Calbuco volcanic plume changed the optical properties of aerosol. For these purposes aeorosol optical properties using AERONET products from Level 2 version data were retrieved.

## 2.3. MODIS

In this study, the Aerosol Optical Depth Level 2 products were used from Moderate Resolution Imaging Spectroradiometer (MODIS) instruments onboard the Aqua Satellite. MODIS measures radiances at 36 spectral bands in wavelengths ranging from 0.4 μm to 14.4 μm using calibrated reflectance data from the following bands, 0.47, 0.55, 0.66, 0.86, 1.24, 1.6, and 2.13 μm. The provided spatial resolution are 250 m by 250 m for 0.66 and 0.86 μm and 500 m by 500 m for 0.47, 0.55, 1.24, 1.6, and 2.13 μm. To retrieve aerosol optical properties from atmosphere, MODIS uses two different and independent algorithms: Deep Blue is responsible for aerosol products of land retrieval only, and Dark Target is a separate algorithm for aerosol products over land and ocean [34,35]. To derive the AOD data from MODIS/AQUA all over South America, the Deep Blue Aerosol Optical Depth layers from the website https://worldview.earthdata.nasa.gov/ were selected. The sensor/algorithm resolution used was 10 km at nadir, imagery resolution of 2 km at nadir, and daily temporal resolution. It was retrieved the AOD product at 550 nm from aerosol version 3-Level 2.

### 2.4. CALIPSO

CALIPSO satellite, which launched in April 2006, flies in a 705 km sun-synchronous polar orbit with an equator-crossing time of about 13:30 local solar time [36]. The primary instrument aboard CALIPSO, and the one used in this study, is the Cloud-Aerosol Lidar with Orthogonal Polarization (CALIOP), working based on two-wavelength laser (532 nm and 1064 nm) with a pulse repetition rate of 20.16 Hz [37]. The CALIOP data products are assembled from the backscattered signals and divided in two categories: level 1 profiles and Level 2 profiles, which are compounded by profile and layer products. Level 1 products are used to derive Level 2 products, which, in turn, are organized into three different types: layer and profile products and the vertical feature mask (VMF) [36].

The set of CALIPSO algorithms uses an aerosol classification scheme to assign each aerosol layer to one of the six aerosol types, namely dust, biomass burning, clean continental, polluted continental, marine, and polluted dust for version 3 data [38]. Version 4 products from CALIPSO data were released in November 2016 and include several improvements concerning the aerosol subtyping and lidar ratio retrieval. The most relevant improvement in Version 4 products algorithm is the possibility to identify aerosol subtypes in the stratosphere [39]. These subtypes are associated to four aerosol types: polar stratospheric aerosol (PSA), volcanic ash, sulfate/other, and smoke. For tropospheric aerosol subtypes, several improvements were made: all aerosol subtypes are retrieved over polar regions; a dust-marine aerosol subtype is introduced, corresponding a mixtures of dust and marine aerosols near the ocean surface; and the polluted continental and smoke subtypes have been renamed polluted continental/smoke and elevated smoke, respectively. In addition, lidar ratio values were revised for clean marine, dust, clean continental, and elevated smoke subtypes [39]. To derive Calbuco's aerosol layer altitude, bottom and top, the AOD and lidar ratio values, the CALIPSO level 2 V4 aerosol data were used. Although version 4 data have been released recently, we preferred to use this newer version since the results and improvements have already been published and approved by peer review evaluation.

Level 1B V4 profile data, obtained from the website https://subset.larc.nasa.gov/calipso/, were used to retrieve 532 nm total attenuated backscatter profile to compare it with SPU lidar data profiles. These data were averaged around 400 km in the closest distance overpass from the SPU lidar station, which, for this case, was 236 km, while, for 23–25 April, was 532 nm. Total attenuated backscatter profiles were averaged over 400, 900 and 400 km, respectively.

### 2.5. The Ozone Mapping and Profiler Suite (OMPS) Limb Profiler (LP)

The OMPS LP is a Limb Scatter (LS) sensor designed to observe the Earth's limb radiance in the 290–1020 nm spectral range. In this region of the spectrum, the solar radiance is the prime source of light being scattered by atmospheric molecules (Rayleigh scatter), suspended liquid and solid particulates (aerosols), clouds and the Earth's surface. The OMPS LP uses the measurements of ultraviolet and visible solar radiation scattered from the earth's limb, combining the advantages of both the backscattered ultraviolet and the visible limb occultation methods [40]. The main goal of the OMPS LP instrument is to conduct the best possible measurement of $O_3$ profiles. The instrument is a triple-slit prism spectrometer that senses the limb radiance and solar irradiance for the wavelength range of 290–1020 nm. Two of the slits are pointing 4.25° (~250 km) on either side of the center slit that points perpendicular to the satellite ground track. This design allows the simultaneous measurement of three profiles of $O_3$ and aerosols. Currently, the retrieval aerosol extinction profiles at 675 nm (Version 1.0) use Chahine nonlinear relaxation algorithm [41], assuming a specified bimodal lognormal size distribution with the coarse-mode fraction tuned to produce a value of the Ångström exponent (525/1020 nm) around 2. Aerosol extinction profiles at 1 km resolution are retrieved in the altitude range ~10–35 km [42]. In the present study, we used version 1.5 still under development [43]. To derive the aerosol extinction at 532 nm from the original aerosol extinction at 675 nm, we used the values of the Ångström exponents in the wavelengths range 694–532 nm derived for the period from July to October 1991, after the Mt. Pinatubo eruption. The Ångström exponents at four height ranges: the tropopause

to 15 km, 15–20 km, 20–25 km and 25–30 km [44,45]. For the conversion from aerosol backscattering to aerosol extinction, we used the conversions factor for the wavelengths range 694–532 nm in the same altitude layers cited above [44,45].The stratospheric aerosol optical depth (SAOD) was calculated for 27 April for both the lidar and the five OMPS LP measurements coincident in a radius of 500 km or less. For a consistent comparison of the SAOD values between the lidar and OMPS LP, it was derived in the altitude range between the tropopause (16.5 km on April 27) and 25.5 km, the highest level of a valid lidar signal. The tropopause altitude was determined from the sounding station Campo de Marte Airport (WMO-83779) located in São Paulo city, at 722 m of altitude conducted at 00:00 and 12:00 UTC. Tropospheric aerosols lidar measurements are conducted at São Paulo at 7.5 m vertical resolution. However, it causes the aerosols profiles retrieved in the upper troposphere and lower stratosphere to be very noisy. Thus, the lidar stratospheric profile was smoothed using the moving means from 90 levels, for comparing the lidar and OMPS LP stratospheric profiles. Cross sections of the aerosol extinction from OMPS LP were created along the satellite trajectories using the profiles from the three slits separately for 26, 27 and 28 April 2015. For the same day, the three cross sections cover a little more than 5°in longitude. The altitude of the tropopause at 12 Z is denoted in the figures by the dashed red line and it is also reported in the left corner of the slit L image each day.

The wind profile from the 12:00 UTC sounding was used to calculate the mean zonal wind in the layer 18.5–20.5 km. For further analysis, mean zonal and meridional wind components from the two neighbor levels at 19.415 and 19.608 km, the only sounding levels located between the base and top of the lidar spike-like measured aerosols layer. Finally, the mean wind vector magnitude and its direction for the layer was determined.

## 3. Results

The current study shows a methodology for assessing the optical properties of aerosol layers from eruption of Calbuco volcano. After three days of Calbuco's eruption, the pyroclastic material and $SO_2$ plumes were advected over 4000 km into the Metropolitan Area of São Paulo (MASP) situated at Southeastern Brazil. MASP is heavily populated and one of largest metropolitan areas in the world, where about 80% of the air pollution in the MASP is attributable to vehicle emissions [46], and the principal contributors for local pollution are heavy-duty vehicles (52%), soil dust (26%), light-duty vehicles (18%), and others (4%), probably includign industrial processes [47]. Conversely, air quality related issues are also attributed to long- and mid-range transport of biomass burning aerosols (BBA), which is frequently observed in MASP during the southern dry season (June to September) [48]. Marine aerosols detection are also common at MASP given its proximity to the sea, about 90 km [49].

In this sense, SPU LALINET lidar station (Lat: 23.56°S, Long: 46.74°W) conducted lidar measurements on 27 April when it could detect a narrow layer of aerosol between around 18 and 19.3 km. The nearest aerosol extinction profile from OMPS LP showed its maximum at the same altitude as the SPU lidar station, with better agreement with the Ångström exponents derived from mid-latitude particle sizes distribution measured during the four month following the Mt. Pinatubo eruption. However, the OMPS LP extinction profile showed the Calbuco particles were present from 16.5 to 22.5 km [50]. Hypothesis were raised to explain that contradictory results, with the spatial inhomogeneity of the aerosol layers being the most plausible. In the present study, we revisited this issue and demonstrated the former hypothesis was the reason for the apparent contradiction. The 532 nm total attenuated backscatter profile retrieved by SPU lidar station also showed a good agreement in consonance with CALIPSO's 532 nm total attenuated backscatter profile from the Calbuco volcanic aerosol plume. The early detection of these plumes and their characterization is of key importance, as due to the circulation pattern there is a decoupling in different layers at distinct altitudes, which were monitored about one-half month later over the southern Africa region [18], and their results for the earlier stages of eruption were compared with our results to better infer the behavior of the plume.

After three days of Calbuco's eruption, the pyroclastic material and $SO_2$ plumes were transported over 4000 km into the metropolitan area of São Paulo. The plume was detected between 18 and 19 km, above sea level. Considering the radiosounding data, this aerosol layer was detected above tropopause, which for São Paulo has a maximum altitude of 16 km above sea level, as shown in the Section 5. Satellite data from CALIPSO and MODIS, as well as sunphotometric input (AERONET), enabled estimating quantities such as plume optical depth and its geometrical properties (top/bottom heights and thickness). The volcanic plume were stationary all day as shown in the temporal distribution curtain plot in Figure 3. The plume was tracked down by CALIOP instrument on board of CALIPSO satellite and shown for 23–25 April for the entire plume spanning over a large area of the continent including Argentinian, Brazilian and Paraguayan territories and extending into the Atlantic Ocean.

As Calbuco's second eruption occurred right around 14:20 UTC, TERRA/MODIS products showed the fist results. Figure 1 shows the AOD (550 nm) product. Following the A-TRAIN constellation, CALIPSO gave the total attenuated backscatter at 532 nm. Three days later, some of the dispersed plume reached MASP, as can be seen by the air mass back-trajectory provided by Hybrid Single Particle Langrangian Integrated Trajectory Model (HYSPLIT) (Figure 1). HYSPLIT trajectory model [51] was used to calculate backward trajectories and derive information on where, when and at which altitude aerosol layers from volcanic eruption were transported. Four-day back-trajectories of air-masses starting at the to SPU lidar station coordinates (Lat: 23.56°S and Long: 46.74°W) were calculated using the GDAS database from Global Data Assimilation System, for four different altitudes ranging from 17 to 20 km above ground level (a.g.l.). The back-trajectories starting at 18 UTC and with altitude level of 17 and 18 km a.g.l. came originally from the same region of the Calbuco eruption, as depicted in Figure 1.

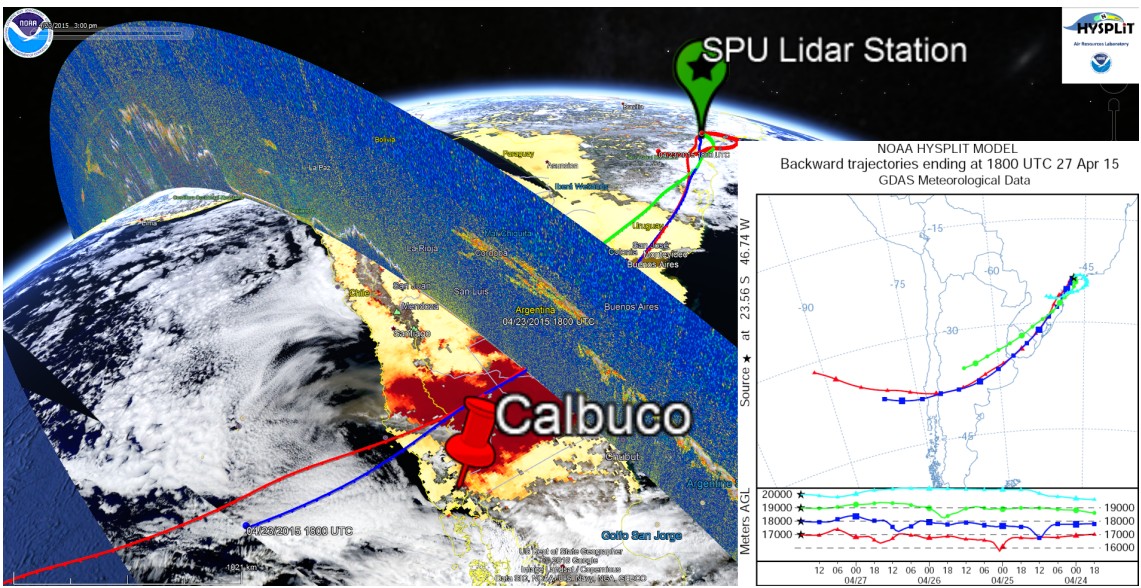

**Figure 1.** Calbuco eruption detected by TERRA/MODIS and CALIPSO satellite on 23 April 2015 and the air masses trajectories from HYSPLIT model pointing to the aerosol plumes travel forward to São Paulo region.

On 23 April, CALIPSO overpass was around 560 km from the volcano and the plumes were between 15 and 19 km, as shown by the total attenuated backscatter profile at 532 nm presented in Figure 2 (left). Using the CALIPSO 5-km aerosol layer version 4 data, we could retrieve an AOD at 532 nm of $0.32 \pm 0.07$ and lidar ratio of $44 \pm 9$ sr. Thus, the aerosol plume was classified as stratospheric layer and volcanic ash subtype. The next volcanic aerosol layer detection was on 24 April as the plume expanded northeasterly, as shown in Figure 2 (middle-left). In this case, CALIPSO satellite detected the stratospheric aerosol plume between 14 and 15 km, 5 km aerosol layer version 4 data presented an AOD value of $0.28 \pm 0.05$ and lidar ratio of $44 \pm 9$ sr, and the aerosol layer was assigned

as stratospheric layer and volcanic ash aerosol subtype. On 25 April , the aerosol plume moved in the northeasterly direction and CALIPSO data show two distinct decoupled layers over the south region of Brazilian territory, between 15 and 17 km, for the first layer and 17 and 18 km for the second one. AOD values were $AOD_{1st} = 0.05 \pm 0.01$ and $AOD_{2nd} = 0.021 \pm 0.007$, for the first and second layers, respectively. Both layers were classified as stratospheric aerosol and sub-classified as sulfate. On 27 April, a volcanic plume was again identified by CALIPSO about 700 km from SPU LALINET station. The layer remained between 18.5 and 19.0 km. In this case, optical properties values retrieved by CALIPSO were $AOD = 0.05 \pm 0.01$ and lidar ratio of $44 \pm 9$ sr. The aerosol layer for this case was classified as stratospheric layer and sub-classified as volcanic ash. Around 12:00 UTC, a layer between 18 and 19 km was for the first time detected by SPU lidar station, matching with CALIPSO's retrieval. Figures 2 and 3 present SPU range corrected backscattered signal at 532 nm and total attenuated backscatter profile retrieved by CALIPSO also at 532 nm. As shown in the figures, both instruments detected the volcanic aerosol layer at the same altitude, between 18.5 and 19 km. According to Figure 2 (right), both attenuated backscatter profiles at 532 nm, retrieved by CALIPSO and SPU station, respectively, were in agreement in intensity and on the physical properties such as top and bottom volcanic layer altitude. The same can be checked in Figure 3 (top), which presents the 532 nm total attenuated backscatter profile for the CALIPSO overpass distance of 700 km from SPU station from the latitude. As shown in Figure 3 (top), there was an aerosol plume detected between 19 and 20 km and between the latitude coordinates of −21.02 and −27.10. The same aerosol plume can be seen in Figure 3 (bottom), which presents the 532 nm range corrected signal retrieved by the SPU lidar station during 12:22 UTC to 21:10 UTC. The aerosol plume was detected at 19 km throughout the entire measurement period. The color bar on the right side of 532 nm total attenuated backscatter profile from CALIPSO satellite and from SPU Lidar station range corrected signal at 532 nm indicate the value of total (molecular and aerosol) backscatter signal retrieved by each system. Both graphics show the coincidence between the volcanic plume detection by both platforms, CALIPSO satellite and SPU lidar ground-based station.

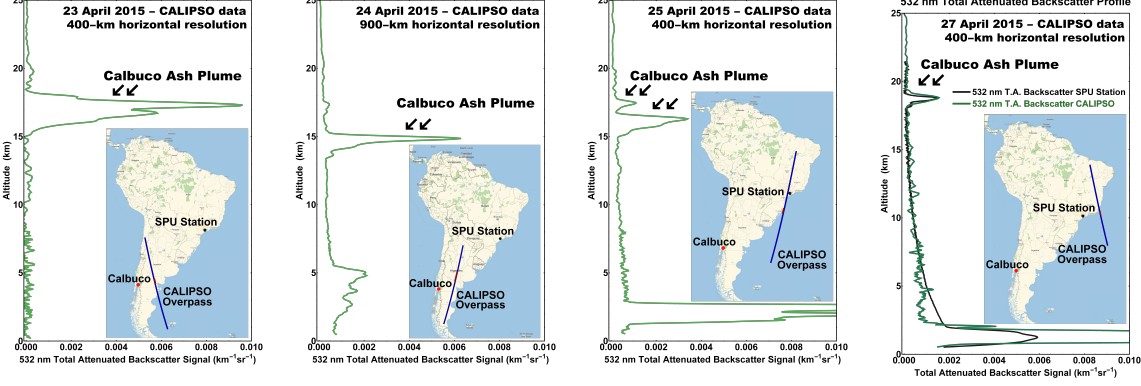

**Figure 2.** CALIPSO overpasses and the Total attenuated backscatter profile at 532 nm retrieved using Level 1B V3 data for consecutive days of Calbuco eruption: (right) the attenuated backscatter profiles at 532 nm retrieved by CALIPSO and SPU station, respectively. They are in total agreement in intensity and physical properties such as top and bottom volcanic layer altitude.

The São Paulo AERONET Station obtained AOD at 355 nm and 532 nm were retrieved during two periods I (25, 26, 28, and 29 April) and II (27 April only), as presented in Table 1. The difference between the two periods provides the volcanic plume detected over the MASP area on 27 April 2015. From the daily evolution, presented in AERONET AOD data, one can see an increase in both AOD wavelengths on 27 and 28 April, which was due to the volcanic material, which covered the region and impacted the optical properties in a distinctive way. Assuming the well mixed atmosphere over the MASP area did not change significantly, one can assume the average increasing of 0.09 in the in the

AOD at 532 nm and 0.12 in the AOD at 355 nm, concerning the average values for the whole period presented in Figure 4. As shown in the figure, the AOD values for 532 nm were lower than 0.1 for the days before the SPU lidar station detection. On 27 April, the AOD increased substantially, achieving values of 0.4 for the end of this day and with mean AOD values of 0.22 ± 0.09. The same pattern was evidenced the 355 nm wavelength, where the AOD values were around 0.15 on the days before the volcanic plume detection at São Paulo, which then increased up to 0.63.

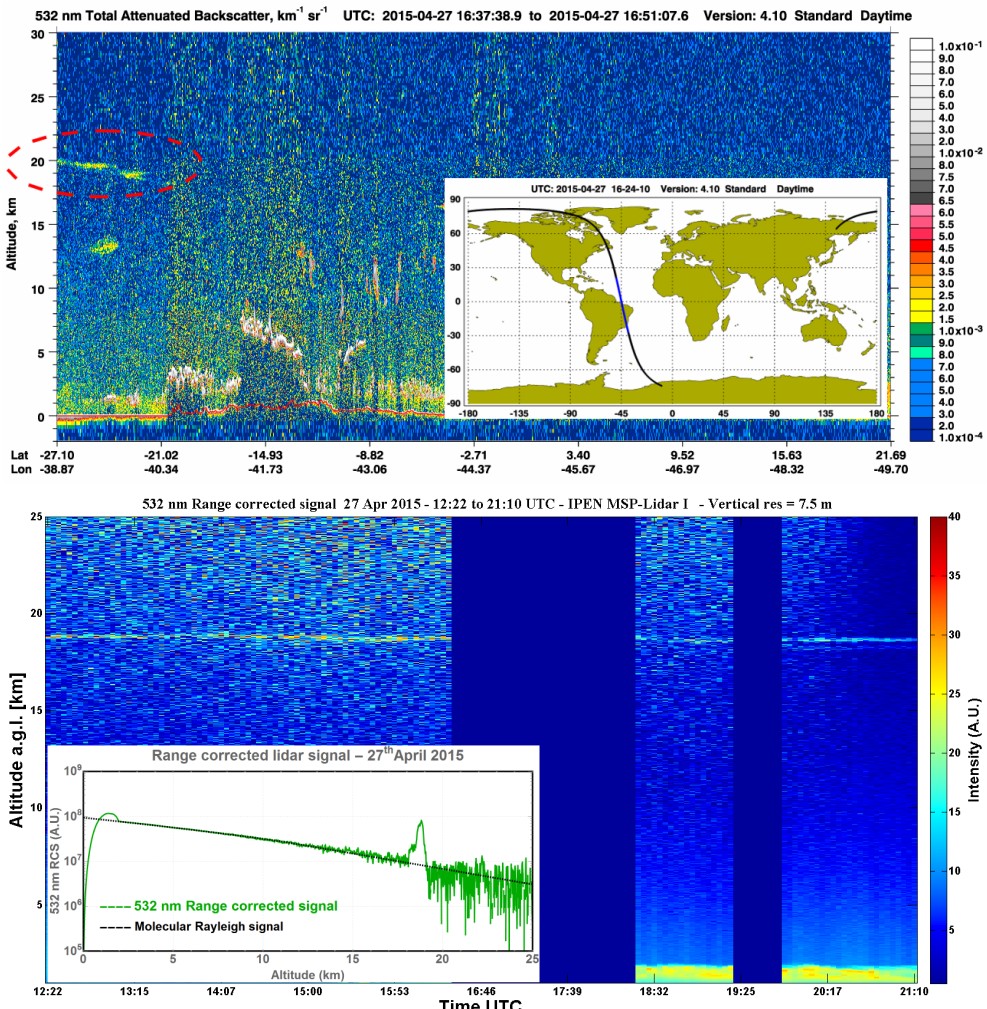

**Figure 3.** Aerosol plume from Calbuco Volcano retrieved by CALIPSO's total attenuated backscatter profile at 532 nm and the SPU Lidar station range corrected signal at 532 nm.

When the aerosol size distribution Almucantar level 1.5 for 26–28 April was investigated, the domination of the fine mode aerosol over coarse mode was found, especially on 27 and 28 April, when the fine mode size distribution increased substantially, accompanying the increasing of the values of AOD values, as presented on Figure 5. The size distribution values spanned from 0.05 to 20 m, which were within the expected values for volcanic ashes [52]. These important results retrieved from AERONET sunphotometer presented on Figures 4 and 5 have the same pattern revealed on the study of long range transport of Eyjafjallajökull volcanic plume over Athens, Greece presented by Kokkalis et al. [53], where low AOD values of approximately 0.1 were retrieved at 500 nm before the volcanic ashes arrived; during the advection of the plume, the AOD peaked at a value of 0.25; and, after the volcanic plume event, the AOD values decreased again to 0.20. In addition, Kokkalis et al. [53] showed the domination of fine mode particles during the volcanic ash detection, where fine-mode

particles inside the atmospheric column were on the order of 59.1–60.9% before and after the event, and increased to 76.8–78.0% during volcanic event detection.

**Table 1.** Daily average AOD at 355 nm and 532 nm retrieved by São Paulo AERONET station and MODIS satellite.

| Date Period | AOD at 355 nm | AOD at 532 nm | AOD at 530 nm (MODIS) |
| --- | --- | --- | --- |
| 25 April | $0.12 \pm 0.03$ | $0.08 \pm 0.01$ | 0.09 |
| 26 April | $0.14 \pm 0.08$ | $0.11 \pm 0.04$ | - |
| 27 April | $0.33 \pm 0.16$ | $0.22 \pm 0.09$ | 0.09 |
| 28 April | $0.34 \pm 0.04$ | $0.23 \pm 0.02$ | 0.11 |
| 29 April | $0.21 \pm 0.08$ | $0.14 \pm 0.06$ | 0.12 |

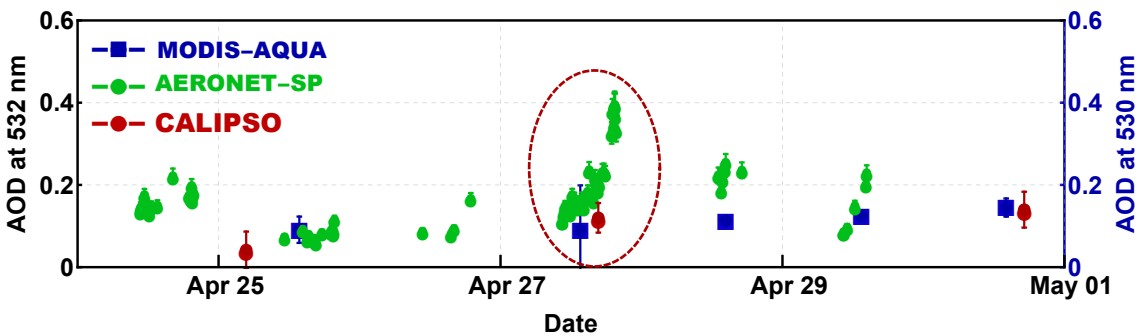

**Figure 4.** AOD evolution at 532 nm from AERONET and CALIPSO and 530 nm from AQUA/MODIS. On 27 April, there was a significant change in AOD values according to the volcanic plume presence, highlighted by the red circle.

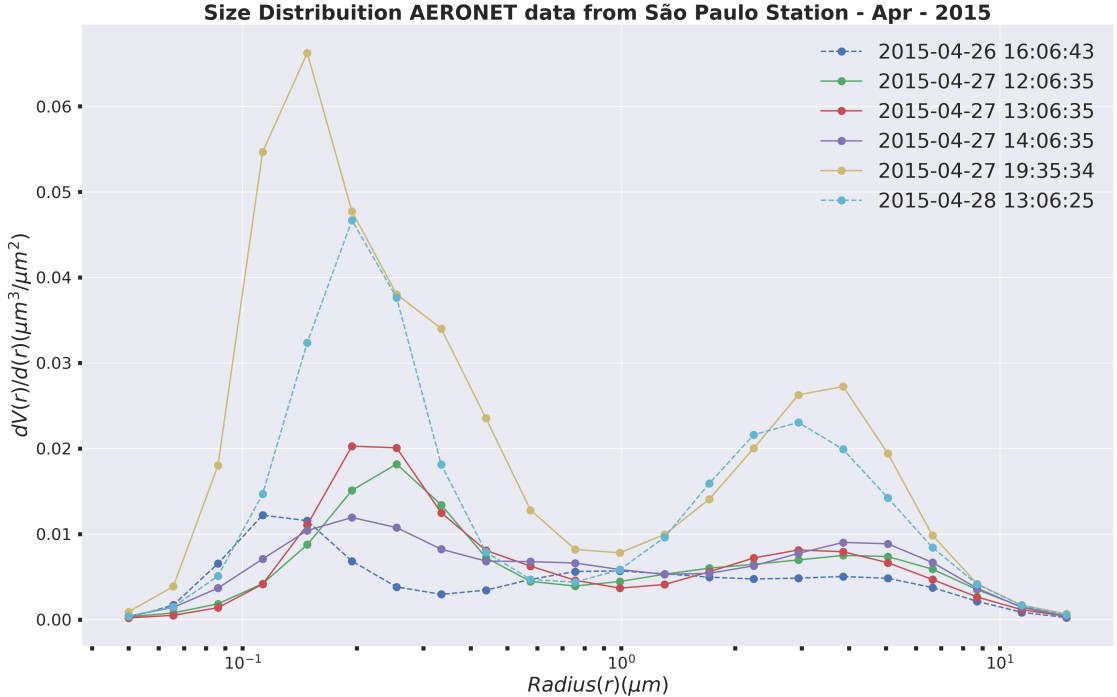

**Figure 5.** AERONET size distribution on 26–28 April 2015.

## 4. The OMPS Limb Profiler Analysis

Another remote sensing instrument used to retrieve the aerosol volcanic plume from Calbuco eruption over São Paulo area was the Ozone Mapping and Profiler Suite Limb Profiler, onboard of Suomi National Polar-orbiting Partnership (Suomi NPP). Table 2 shows the OMPS LP measurements on April 27 coincident inside a radius of 500 km from the São Paulo lidar station. All measurements were conducted by the right slit of the OMPS LP during its overpass east of the SPU lidar station. The SAOD at the first measurement in the table, south of the station, shows the lowest SAOD of the five, increasing monotonically its value in the following three as the OMPS LP moves north. All the time and space coincident SAOD measurements were an order of magnitude lower than the SAOD measured by the SPU lidar station, 0.159.

**Table 2.** OMPS LP measurements coincident in a radius of 500 km around the SPU lidar station during 27 April 2015, when the Calbuco's volcanic plume was detected by over São Paulo.

| Time (UTC) | Latitude | Longitude | Distance | SAOD |
|---|---|---|---|---|
| 16:26:40 | 24.9°S | 42.5°W | 458 km | 0.031 |
| 16:26:59 | 23.8°S | 42.7°W | 408 km | 0.041 |
| 16:27:27 | 22.7°S | 43.0°W | 393 km | 0.066 |
| 16:27:36 | 21.6°S | 43.3°W | 417 km | 0.053 |
| 16:27:55 | 20.5°S | 43.5°W | 473 km | 0.040 |

The profiles of the five OMPS LP measurements are shown in Figure 6 together with the smoothed lidar profile. The extinction coefficient profile retrieved by the SPU lidar station matched the dual aerosol layer structure shown in all OMPS LP profiles except in the one located north of the lidar. In the lidar extinction coefficient profile, the upper aerosol layer showed equal or higher magnitudes of the aerosol extinction than the OMPS LP at the same level. In contrast, in the lower layer, the lidar retrieval showed a spike-like profile around 19 km with ±1 km vertical extension, while all OMPS LP profiles showed the aerosol layer located from around 10 to 21 km. In addition, the magnitude of the aerosol extinction at the altitude of the aerosol layer measured by the lidar, around 19 km was larger in all OMPS LP profiles.

Figure 7 shows the cross sections of the aerosol extinction along the trajectories using the profiles from the three slits separately for 26–28 April 2015. The altitude of the tropopause at 12 Z is denoted in the figures by the dashed red line and it is also reported in the left corner of the slit L image on each day.

The evolution of the core of the stratospheric aerosol is very interesting: on 26 April, it was located mainly below the tropopause altitude at the three longitudes, where the OMPS LP measurements took place that day.

Table 2 shows the SAOD derived from the individual profiles measured by OMPS LP on 27 April 2015 located in a radius equal or lower than 500 km from the São Paulo lidar station (23.56°S, 46.74°W). That day, the orbit of the OMPS LP passed east to the station and the measurements conducted by the instrument left slit (which were used to generated the cross-section identified with the mean longitude 37°W) was the eastern among the other two also located east of the lidar but at the mean longitudes of 39°W and 42°W. These measurements at 39°W and 42°W showed the core of layer well below the tropopause. The cross sections from 28 April showed the core of the layer well below the tropopause. The double cores on 26 April (41°W and 46°W) also were present on 28 April (32°W and 37°W). In particular, the patterns from 26 April at 46°W and 28 April at 37°W were very similar but, on this last day of measurements, the magnitude of the cores decreased as well as their altitude. Similar behavior was found when 26 April at 44°W and 28 April at 34°W were compared with 26 April at 41°W and 28 April at 32°W.

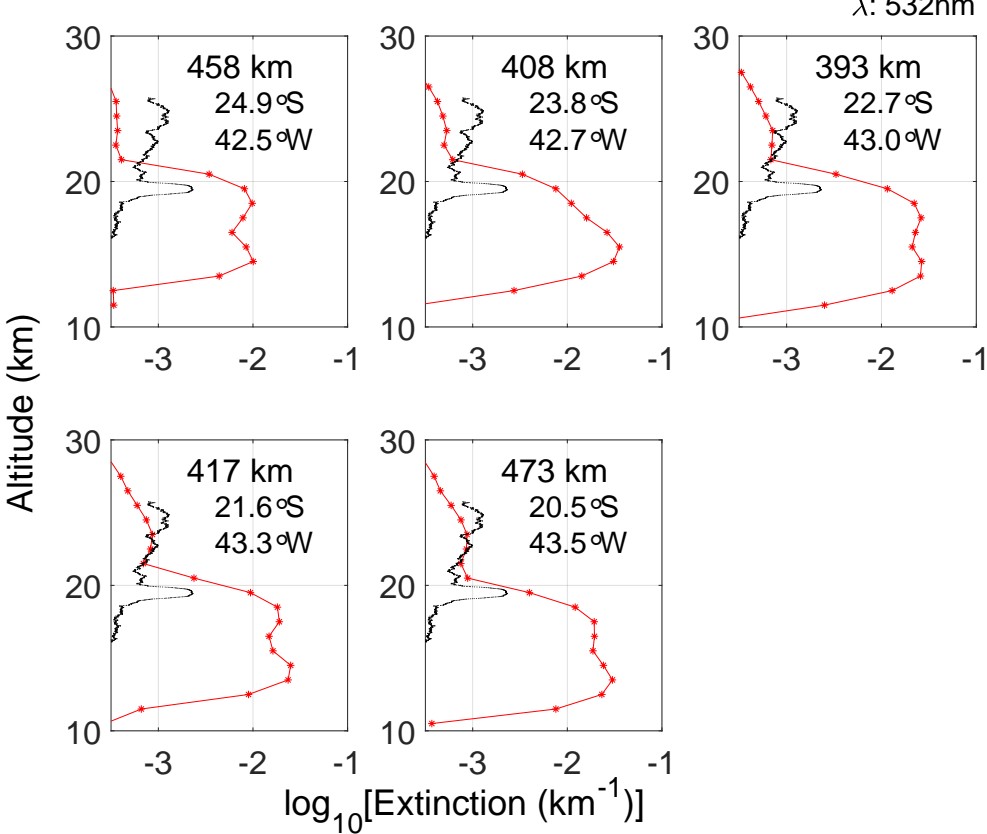

**Figure 6.** Aerosol extinction profiles of the five OMPS LP measurements coincident in time and space (500 km radius around the SPU lidar station) are shown in red. The extinction coefficient profile retrieved from measurements of 27 April 2015 using the Klett–Fernald–Sasano inversion formalism, in black, is also shown.

The cross section on 27 April from the right slit at $42°$W, the nearest longitudinally to the lidar site, showed at $24°$S aerosol extinction magnitudes well below 0.01 km$^{-1}$ throughout the entire layer. That feature explains the low values of the aerosol extinction below 18 km, but does not explain the spike-like layer from around 18 to 20 km. The cross section from the center slit the same day at $39°$W showed a local maximum below 20 km at $24°$S that matches altitude of the spike-like layer present on the lidar extinction profile.

The distances, considering the same latitude of $\sim 24°$S, between the longitudes of the location of the lidar ($\sim 47°$W) and the cross sections at $42°$W and $39°$W were, respectively, $\sim$530 and 810 km. The mean zonal wind of the layer from 18.5 to 20.5 km at 12:00 UTC was $-12.9$ km/h, noting that the maximum speed in the layer was $-21.4$ km/h. The lidar profile was measured at 12:00 UTC and the OMPS LP measurements were taken approximately at 16:30 UTC with a difference of 4.5 h. Then, rough estimates of the zonal transport of the lidar aerosols spike-like layer measured at $46.7°$W produced 60 km when using the mean zonal wind and 100 km when using the maximum zonal wind of the layer. Those results allowed locating the spike-like layer measured by the lidar at around $45.5°$W at 16:26 UTC, the time the OMPS LP measurements were conducted. The nearest cross section that day, at $42°$W, showed at $24°$S a broad layer extending from right below 20 km down to around 15 km, most of it not present in the lidar spike-like layer. Although the aerosol extinction of the layer shown in the OMPS LP cross section matched the magnitude of aerosol extinction in the lidar measured layer, they did not agree in their vertical extension.

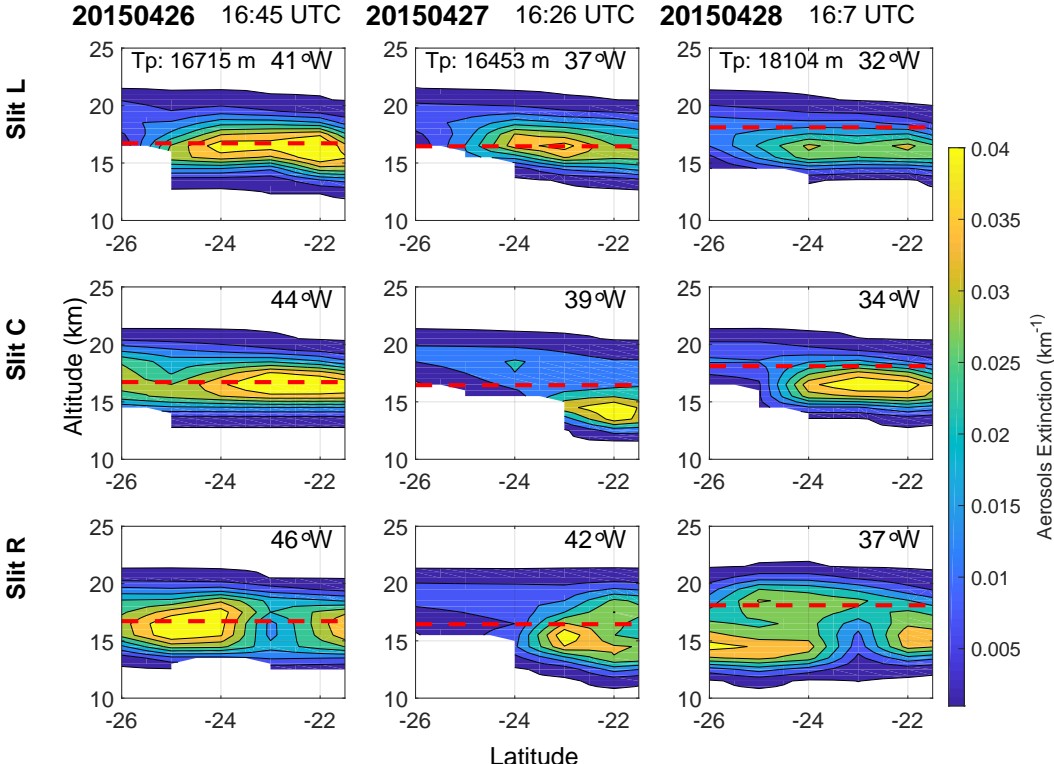

**Figure 7.** Cross sections of the OMPS LP aerosol extinction along the satellite trajectories. Profiles from the three slits were used individually, belonging to 26–28 April 2015. The tropopause altitude at 12 Z is denoted by a dashed red line. Tropopause altitude is also reported in the left corner of the slit L image each day.

Then, a more precise and detailed transport analysis was conducted using the W–E and S–N components from the two sounding levels located between the base and top of the spike-like lidar measured layer. The magnitude of the mean wind of the layer was 19.4 km/h and its direction 167.1 degrees. Then, at 16:26 UTC, the spike-like layer was located around a degree in latitude south of the station, approximately 25°S. In the 42°W OMPS LP cross section that day, at 25°S the layer was narrow, with its top below 20 km and its base above 18 km. The former analysis helped explain the apparent disagreement between the lidar and OMPS LP measurements, regarding the vertical extension of the layer.

## 5. SPU Lidar Analysis and Transmission Method

All the results presented, such as CALIPSO total attenuated backscatter profile and AOD at 532 nm, AQUA/MODIS AOD at 530 nm, the AOD and aerosol size distribuition from AERONET sunphotometer, the extinction profile retrieved by OMPS LP, as well as range corrected signal at 532 nm from SPU lidar station, corroborated for the volcanic aerosol plume detected and transported from the Calbuco Volcano to São Paulo metropolitan area. The most important question in the case of an aerosol layer detected at the lower stratospheric region using a lidar system is regarding the capability and advantage of retrieving optical and physical information of the aerosol plume with a high temporal and vertical resolution. Using the raw backscatter lidar signal and applying the formalism developed by Platt [27], it is possible to calculate the transmittance of the volcanic aerosol layers, the aerosol

optical depth and lidar ratio, which can indicate the plume compounds and type. The raw lidar backscatter signal is defined by the lidar equation [54] as follows:

$$P(\lambda, z) = P_o \frac{c\tau}{2} A \eta O(\lambda, z) \frac{\beta_m(\lambda, z) + \beta_{aer}(\lambda, z)}{z^2} \, exp\left[-2\int_o^z \alpha(\lambda, z')dz'\right] \tag{1}$$

where $P(\lambda, z)$ is the power signal detected at distance $z$, $z$ is the distance of the atmospheric volume investigated, $P_o$ is the power emitted by the laser source, $c$ is the speed of light, $\tau$ is the laser pulse duration, $A$ is the effective area of the telescope receptor, $\eta$ is a variable related to the efficiency of the lidar system and $O(\lambda, z)$ is the laser-beam receiver-field-of-view overlap function. The most important quantities are the backscatter $\beta(\lambda, z)$ and the extinction $\alpha(\lambda, z)$ coefficients, which refer to both the aerosol and molecular contribution. Using the aerosol and molecular backscatter coefficient, it is possible to derive the backscatter ratio (Equation (2)):

$$R(\lambda, z) = \frac{\beta_{aer}(\lambda, z) + \beta_{mol}(\lambda, z)}{\beta_{mol}(\lambda, z)} \tag{2}$$

where $\beta_{aer}(\lambda, z)$ is the backscatter coefficient compound due the aerosol and $\beta_{mol}(\lambda, z)$ is the backscatter coefficient portion due molecular contribution on the atmosphere. Applying the formalism discussed by Bucholtz [55], the molecular backscatter and extinction profiles of atmosphere are calculated using the radiosonde launched twice a day from the weather observation base SBMT Marte Civ/Mil (Lat: 23.52°S and Long: 46.63°W), which is around 10 km of distance from the SPU Lidar station. Radiosonde data are also used to calculate the end of the troposphere and the beginning of the lower stratosphere. According to Equation (2), for pure Rayleigh atmosphere condition, where there are no aerosol signal, $R = 1$; then any value of R greater than 1 is an indication of aerosol layers presented at the atmosphere. To determine the optical depth and the transmission of the aerosol layer detected at the lower stratosphere by the SPU lidar station, the same idea of optical depth and the two-way transmittance calculation for Cirrus Clouds [56] is applied, where, considering single-scattering conditions, both quantities are defined by,

$$\tau(\lambda, z) = \int_{z_{base}}^{z_{top}} \alpha(\lambda, z')dz' \; , \tag{3}$$

$$T^2(\lambda, z) = exp\left[-2\int_{z_{base}}^{z_{top}} \alpha(\lambda, z')dz'\right] = exp\left[-2\tau(\lambda, z)\right] \tag{4}$$

where $z_{base}$ and $z_{top}$ represent the base and the top heights of the aerosol layer, respectively. For atmospheric scenes when lidar detect elevated aerosol layers within the clear-air region, it is possible to calculate the two-way transmittance with reliable accuracy considering the pure molecular signal at the region immediately above and below the aerosol layer [57,58]. To calculate the transmission term, the range corrected lidar signal is adjusted to a simulated lidar signal for the molecular clear-air conditions using the radiosonde data and Bucholtz's formalism [55]. The fit adjustment is applied to the region below the aerosol layer and the region above it. Using this approach, the transmission term for the aerosol layer can be calculated by the following expression

$$T_{fit} = \left[\frac{P_{top}(z)\, z^2}{P_{base}(z)\, z^2}\right]^{\frac{1}{2}} \tag{5}$$

where $P_{top}(z)\, z^2$ and $P_{base}(z)\, z^2$ are the so-called range corrected lidar signal for the top and the base height of the aerosol layer, respectively. Using Equations (4) and (5), it is possible to derive the aerosol optical depth $\tau(\lambda, z)$ for the volcanic layer.

As presented by Platt [27], the lidar ratio can be derived by the integrated form using the integrated backscatter lidar signal from the top to the base of the aerosol layer. The lidar ratio can be found using Equation (6):

$$S(\lambda, z) \ = \ \frac{1 \ - \ \exp\left[-2\,\eta\,\tau(\lambda, z)\right]}{2\,\eta\,\gamma(\lambda, z)} \tag{6}$$

where $\gamma(\lambda, z)$ is the integrated backscatter signal retrieved applying the Klett–Fernald–Sasano analytical method [28–30,59], represented by Equation (7), and $\eta$ is the multiple scattering factor, which can be estimated by Equation (8) [56].

$$\gamma \ = \ \int_{z_{base}}^{z_{top}} \beta(\lambda, \ z')\mathrm{d}z' \tag{7}$$

$$\eta \ = \ \frac{\tau(\lambda, z)}{\exp\left[\tau(\lambda, z)\right] \ - \ 1} \tag{8}$$

Figure 8 presents the vertical profiles of backscatter ratio retrieved by Equation (2) four days after the eruption of Calbuco volcano. This first volcanic aerosol plume detected by the SPU lidar station on 27 April 2015 was observed by our lidar system during the whole day (Figure 3). The volcanic aerosol peak was detected above the tropopause, at the lower stratosphere, which starts around 16 km at São Paulo atmosphere, determined based on the radio sounding temperature profile data presented in Figure 9A. The peak value of the backscatter ratio was 9.5 at 18.8 km, according Figure 8. The thickness of the layer was around 1.3 km; the bottom and the top of the plume were 18 and 19.3 km, respectively. Applying Equation (5) and using the range corrected signal retrieved at the bottom and top of the plume, we could retrieve the layer transmittance term of 0.852. By using Equation (4), we retrieved a SAOD value of 0.159. Combining the AOD of the plume and the backscatter integrated value obtained by the lidar signal between the range altitude of 18 and 19.3 km, as presented in Figure 9B, we retrieved a lidar ratio value of 76 ± 27 sr using Equations (6) and (7). Applying same methodology using retrieved signal from 355 nm channel, the lidar ratio retrieved was 63 ± 21 sr. We could also calculate the Ångström exponent for the volcanic plume as 0.61 ± 0.58.

This is the most important result retrieved by the detection of aerosol plume from the eruption of Calbuco volcano over South America. Several studies have investigated the optical properties of volcanic aerosol layers using lidar systems; these studies resulted in a large range of lidar ratio values. Wang et al. [60] studied the optical properties of aerosol layers from 1 to 6 km detected using a lidar system in southern Europe during the Etna eruptions periods. Lidar ratio values ranging from 47 to 63 sr were retrieved using the wavelength of 355 nm, and backscatter related Ångström exponent ranging from 0.9 to 2.8, suggested the volcanic layers detected were a mixture of sulfates and sub-micrometric ash particles. Hoffmann et al. [61] investigated stratospheric aerosol plumes loaded in the atmosphere by the eruption of the Kasatochi volcano in 2008 using an aerosol Raman Lidar and a Micro Pulse Lidar (MPL). In this study, volcanic aerosol plumes were detected between 11 and 18 km and the lidar ratio values retrieved at 355 and 532 nm were 63 ± 10 sr and 76 ± 10 sr, respectively. These lidar ratio values was attributed not only to sulfates but also with a mixture of absorbing components. In 2010, Mattis et al. [62] presented a completed study of optical and microphysical properties of aerosol layers detected in the upper troposphere and in the lower stratosphere in the framework of EARLINET. Lidar ratio values ranging 30–60 sr for the wavelength of 355 nm and 30–45 sr for the wavelength of 532 nm were retrieved using a multi-wavelength aerosol Raman lidar. The Backscatter-related Ångström exponents ranged from 1.0 to 2.0. This study shows layers compounded by absorbing and spherical aerosol particles. Several other studies on the framework of EARLINET investigated and classified aerosol optical properties of volcanic ashes. Ansmann et al. [63] showed optical and microphysical properties of aerosol plume from Eyjafjallajökull eruptions in southern Iceland detected over Leipzig and Munich (Germany) using lidar systems. The lidar ratio values ranged from 55 ± 5 sr (Munich) to 60 ± 5 sr (Leipzig) at 355 and 532 nm in the main ash layer

were retrieved, as well as ash mass concentrations on the order of $1000 \pm 350$ mg/m$^3$. The retrieved Ångström exponent was $0.03 \pm 0.40$. In this study, the authors classified the aerosol plume as ash and pure dry ash, over Leipzig and Munich, respectively.

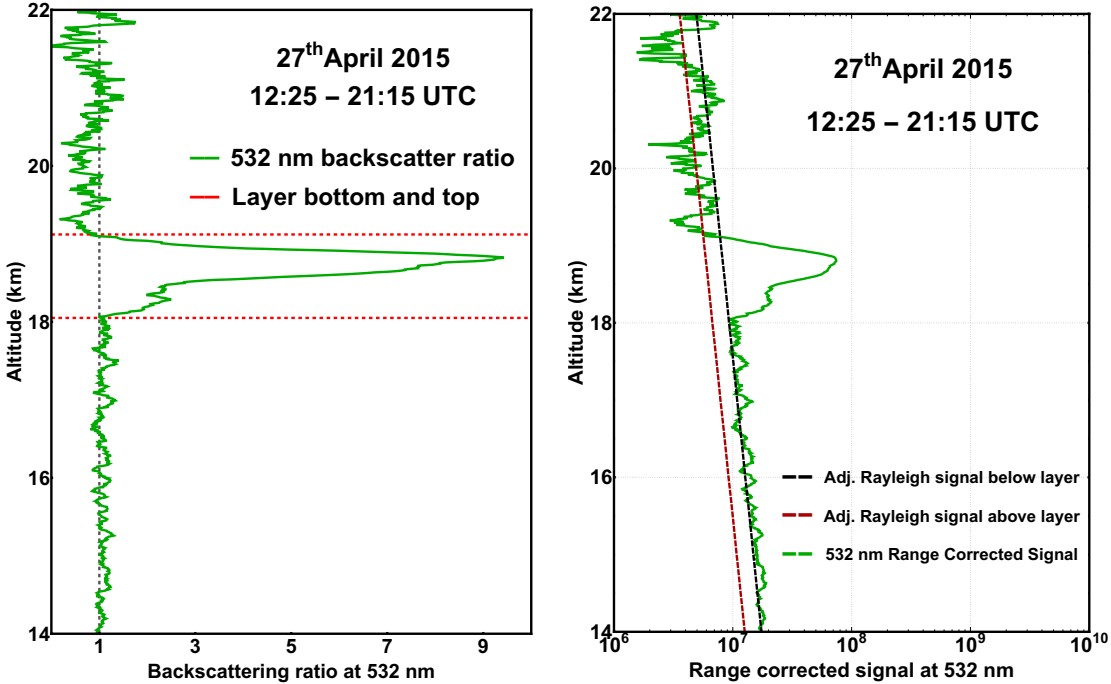

**Figure 8.** The 532 nm backscattering ratio showing the bottom and top altitude of the aerosol plume (**left**); and SPU Lidar Range Corrected Signal at 532 nm together with the molecular Rayleigh adjustment retrieved on 27 April 2015 (**right**). The transmission method was applied to calculate optical transmission of the volcanic plume, and therefore, estimate AOD and Lidar Ratio value.

Ansmann et al. [64] and Groß et al. [65] also detected volcanic aerosol classified as pure dry ashes detected around 2.6–3.5 km and with lidar ratio values of 50–60 at 355 nm and $55 \pm 5$ sr for the 532 nm. The Ångström exponent ranged from $1.1 \pm 0.4$ to $1.5 \pm 0.1$, suggesting larger particles in the upper layer. With all these results, the volcanic aerosol layers were classified as pure dry ashes. Due the Eyjafjallajökull's eruption in April–May 2010, several EARLINET lidar station detected volcanic plumes all over Europe and several studies could retrieve optical and physical properties from volcanic plumes. Mona et al. [66] detected volcanic plumes at different altitudes and on different days at Potenza, Italy. Plumes were detected between 2 and 3 km during their first arrival at Potenza and classified as sulfates with some ash mixture. The lidar ratio values for this case were $42 \pm 2$ and $50 \pm 3$ sr for 355 and 532 nm, respectively, and the backscatter-related Ångström exponent ranged from $1.03 \pm 0.07$ to $1.4 \pm 0.3$. After some days, on 13–14 May, another volcanic plume was detected; however, during this period, aerosol dust transported from Saharan desert region was also detected in the atmosphere, at different altitudes. The aerosol plume detected between 1.5 and 2.3 km was also classified as sulfates with some ash particles, with lidar ratio assigned as $60 \pm 11$ sr for the 355 nm wavelength and $78 \pm 12$ sr for 532 nm. Similar to this study and for the same volcanic eruption, Kokkalis et al. [53] revealed the optical and microphysical properties along with mass concentration of the observed volcanic plume after four days of travel over Athens, Greece. In this study, aerosol plumes detected between 1 and 2.3 km presented lidar ratio values of $80 \pm 1$ and $77 \pm 3$ sr for 355 and 532 nm, respectively. Volcanic plumes between 2.5 and 3 km presented lidar ratio values of $78 \pm 1$ and $76 \pm 5$ sr for 355 and 532 nm, respectively; and high plumes detected at 5–6 km presented lidar ratio values of $60 \pm 2$ and $44 \pm 8$ sr for 355 and 532 nm, respectively.

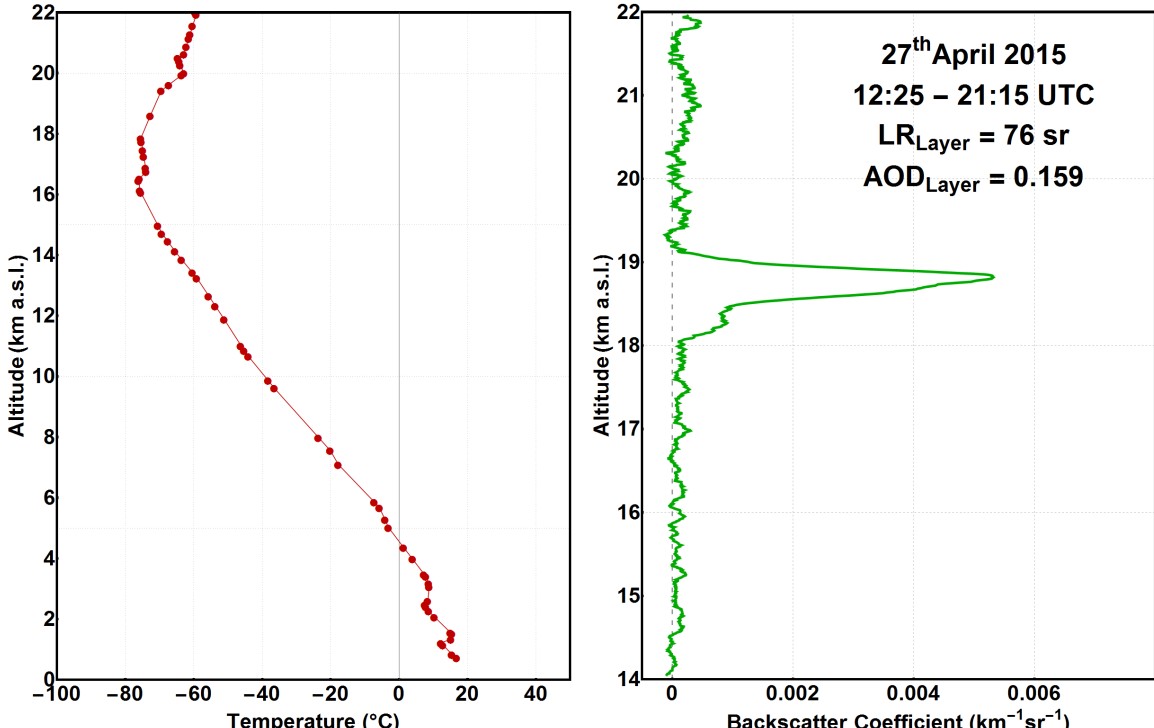

**Figure 9.** Atmospheric temperature profile retrieved by Radiosounding from SBMT Marte Civ Observations on 27 of April 2015 at 12:00 UTC shows the beginning of the lower stratosphere (**A**); and the backscatter coefficient profile at 532 nm retrieved on 27 April 2015 using the Klett–Fernald–Sasano inversion formalism (**B**).

Sicard et al. [67] also published a study of optical properties of volcanic ashes plumes based on detection using a lidar system in different cities of Europe. For this case, the aerosol plumes were detected between 2.7 and 3.7 km and classified as fresh volcanic particles with lidar ratio values of $39 \pm 10$ sr for the 355 nm channel and $32 \pm 4$ sr for 532 nm channel, and the backscatter-related Ångström exponent was $0.68 \pm 0.63$. For Granada EARLINET station, the retrieved lidar ratio values were $47 \pm 7$ sr and $48 \pm 16$ sr for 355 and 532 nm, respectively, and Ångström exponent values ranged from 1.2 to 1.3. In addition, for Granada's EARLINET station, the aerosol plume was classified as sulfates with some ash and the same results were demonstrated by Navas-Guzmán et al. [68]. The volcanic plumes detected by SPU lidar station between 18 and 19.3 km allowed obtaining lidar ratio values of the aerosol layer applying the transmittance method. Lidar ratio of $63 \pm 21$ sr and $76 \pm 27$ sr for 355 and 532 nm, respectively, and Ångström exponent of $0.61 \pm 0.58$ were retrieved by SPU Lidar station over São Paulo. Considering the values of lidar ratio parameter and the Ångström exponent presented in the literature, this volcanic aerosol layer can be classified as sulfates with some ash type. Table 3 summarizes all results of geometrical and optical properties, i.e., aerosol optical depth, lidar ratio, and Ångström exponent, retrieved from EARLINET lidar stations in the specific period of the Eyjafjallajökull volcanic eruption over the European continent.

Recently, Bègue et al. [18] investigatee the space-time evolutions of aerosol plume and $SO_2$ injected at the upper troposphere and lower stratosphere by Calbuco's eruption on 2015 combining satellites instruments (CALIOP, IASI, and OMPS), in situ aerosol counting (LOAC OPC) and lidar observations, and the MIMOSA advection model. The volcanic aerosol layer was detected from 18 to 21 km during the May–July period over Reunion Island site ($21°S$, $55.5°E$). In this study, CALIOP cross section of the 532 nm total attenuated backscatter values of $1 \times 10^{-3}$ to $5 \times 10^{-3}$ $km^{-1}sr^{-1}$ could be attributed to volcanic aerosol particles injected up to the lower stratosphere by the Calbuco. Latitude–altitude cross sections of the scattering ratio retrieved by CALIOP for 16-day selected periods, from 1 to 16 May 2015, show the aerosol plume ranging 17–25 km near to the region of the SPU lidar, which definitely agrees

with the results presented here, where SPU lidar station detected the volcanic aerosol plume from 18 to 19.3 km. However, results produced by MIMOSA model show that air masses containing volcanic aerosols could not move beyond the south region of Brazil due the presence of a subtropical barrier, not specifying the constrained area in this region. SPU lidar 532 nm scattering ratio profile and 532 nm range corrected signal, presented in Figure 8, show that Calbuco's volcanic aerosol plume reached São Paulo Metropolitan Area, which are in the southern part of Brazil.

**Table 3.** Values of geometrical and optical properties, such as aerosol optical depth, lidar ratio, and Ångström exponent, retrieved from EARLINET lidar stations in the specific period of the Eyjafjallajökull volcanic eruption over the European continent and the optical and physical properties retrieved by this study.

| Location | Layer Height (km) | AOD at 532 nm | LR (532nm) | LR (355nm) | Ångström 355/532 | Type | Reference |
|---|---|---|---|---|---|---|---|
| Leipzig | 2.6–4.3 | 0.35 | $60 \pm 5$ | $60 \pm 5$ | $0.03 \pm 0.40$ | Ash | [63] |
| Munich | 2.6–3.5 | - | 50–60 | $50 \pm 5$ | $-0.11 \pm 0.18$ | Pure dry ash | [63–65] |
| Potenza | 2.0–3.0 | - | $42 \pm 2$ | $50 \pm 3$ | $1.4 \pm 0.2$ | Sulfates with some ash | [66] |
| Evora | 2.7–3.7 | 0.07 | $39 \pm 2$ | $32 \pm 4$ | $0.68 \pm 0.63$ | Fresh volcanic particles | [67] |
| Granada | 2.6–2.9 | - | $47 \pm 7$ | $48 \pm 16$ | $0.066 \pm 0.005$ | | [67,68] |
| Cabauw | 2.7–6.0 | 0.53 | $42 \pm 1$ | $44 \pm 24$ | $0.30 \pm 0.03$ | Sulfate-ash mixture | [64] |
| Athens | 3.0–4.8 | 0.05 | $67 \pm 13$ | $89 \pm 3$ | $0.57 \pm 0.26$ | Aged ash/sulfates | [69] |
| Athens | 2.5–3.0 | 0.04 | $76 \pm 5$ | $78 \pm 3$ | $1.72 \pm 0.06$ | Sulfates | [53] |
| Brazil | 18–19.3 | 0.16 | $76 \pm 27$ | $63 \pm 21$ | $0.61 \pm 0.58$ | Sulfates with some ash | this study |

## 6. Conclusions

SPU LALINET lidar station detected at the lower stratosphere region, between 18 and 19.5 km, a volcanic aerosol plume from the eruption of Calbuco volcano on 27 April 2015. Aerosol geometric and optical properties was analyzed applying several remote sensing platforms, such as satellite and ground-based system. CALIPSO total attenuated backscatter profile at 532 nm retrieved using Level 1B V3 data for consecutive days of Calbuco eruption showed the advection of volcanic plume crossing Argentina and Uruguay, and passing over São Paulo on 27 April, which was corroborated by HYSPLIT air mass trajectories at altitude region of 17–20 km. Total attenuated backscatter profile at 532 nm retrieved from CALIPSO data are in good agreement, in terms of altitude and magnitude, with the range corrected signal retrieved by the SPU lidar station.

AERONET Sunphotometer data analysis for the period from 24 April to 1 May showed a substantial increasing on the 532 nm AOD values during the day on 27 April, the same day when SPU lidar station detected the volcanic plume over São Paulo. AERONET size distribution product showed the domination of the fine mode aerosol over coarse mode, especially for 27 and 28 April, which can be attributed to the presence of sulfates.

The space and time coincident aerosol extinction profiles from SPU lidar station and OMPS LP from the Calbuco eruption conducted on 27 April agree in the double layer structure. However, there is an apparent disagreement between the vertical structure of the individual OMPS LP stratospheric aerosol profiles coincident in space and time with the one measured by the São Paulo lidar. The apparent disagreement between vertical extension of the lower layer was demonstrated to be originated by the latitudinal inhomogeneity of the aerosol layer. The SPU lidar station sampled a region of the aerosol layer with low aerosol content from the bottom of the layer until around 18 km, followed in altitude by a patch of aerosol between 18 and 20 km. This profile coincides in the vertical with the local maximum shown at the same latitude and altitude. The analysis of the lower stratospheric zonal winds at the same altitude of the layer allowed finding the time coincidence with the cross section cited above.

Through the application of the formalism developed by Platt [27], it was possible to calculate the transmittance term of the volcanic aerosol layers, the layer aerosol optical depth and its lidar ratio without any assumption. Applying the transmittance method, the AOD value of 0.159 was retrieved, as well as the bottom and top altitude of the volcanic aerosol layer, 18 and 19.3 km, respectively. Combining the AOD of the plume and the backscatter integrated value obtained by the integration

of the backscatter signal within the altitude range of 18–19.3 km, a lidar ratio value of $76 \pm 27$ sr was retrieved for 532 nm. Applying same methodology using retrieved signal from 355 nm channel, the lidar ratio obtained was $63 \pm 21$ sr. For the same plume, an Ångström exponent value of $0.61 \pm 0.58$ was obtained. When comparing these values with those presented in the literature, it was possible to classify the volcanic aerosol layer detected over São Paulo as sulfates with some ash type.

**Author Contributions:** This paper received the individual contribution according the following statement, conceptualization by F.J.S.L. and Eduardo Landulfo; methodology by F.J.S.L. and J.C.A.M.; software by F.J.S.L., J.J.S. and J.C.A.M.; validation, F.J.S.L., J.C.A.M. and G.T.; formal analysis, investigation, writing—original draft preparation and writing—review and editing by F.J.S.L. and J.C.A.M.; supervision, project administration, funding acquisition, E.L.

**Funding:** This research was funded by São Paulo Research Foundation-FAPESP grant numbers 2018/06720-9 and 2015/12793-0; by the National Council for Scientific and Technological Development-CNPQ, grant numbers 152156/2018-6, 432515/2018-6 and 150716/2017-6.

**Acknowledgments:** The authors would like to thank The São Paulo Research Foundation-FAPESP, the National Council for Scientific and Technological Development-CNPQ, the Center for Lasers and Applications (CLA) and the Nuclear and Energy Research Institute (IPEN) for the research support.

**Conflicts of Interest:** The authors declare no conflict of interest.

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
