# Peer review of "Synergetic Aerosol Layer Observation After the 2015 Calbuco Volcanic Eruption Event"

_remotesensing, doi:10.3390/rs11020195_

Round 1
Reviewer 1 Report
Review of "Sinergetic Aerosol Layer Observation after the 2015 Calbuco Volcanic Eruption Event" by Lopes et al., 2018. The submitted manuscript main objective is to optically characterize the Calbuco aerosol plume erupted in 2015. The observation is very precious, being the quantitative remote sensing observations of volcanic plumes very scarce. Moreover, volcanic emissions pose a threat both to air traffic and human health. The structure of the paper overall makes sense, but the narrative is suffering of some hasty writing, references are not adequate and english should be revised across the whole document.
The specific comments can be found in the specific file, while the main issue with the methodology is that some sensors sound very different volumes, due to the orbit and to the very coarse spatial resolution. What is it the added value of using for example OMPS Limb profiler for example? Is the goal of the paper to study the optical properties of the plume from eruption to SP area? Or just to retrieve the optical properties at SP and then validate the observations with satellite sensors? As It is, those two objectives seem to mix up.
Refereces in some sections are sometimes scarce, while in other sections are redundant.

Author Response
Dear MDPI/RS editors and reviewer #1
We would like to express our gratitude for the carefully made review. Our comments and responses are given below:
Review of "Sinergetic Aerosol Layer Observation after the 2015 Calbuco Volcanic Eruption Event" by Lopes et al., 2018. The submitted manuscript main objective is to optically characterize the Calbuco aerosol plume erupted in 2015. The observation is very precious, being the quantitative remote sensing observations of volcanic plumes very scarce. Moreover, volcanic emissions pose a threat both to air traffic and human health. The structure of the paper overall makes sense, but the narrative is suffering of some hasty writing, references are not adequate and english should be revised across the whole document.
Response: We would like to thank the reviewer #1 for the valuable comments. All of them were very important to improve the quality of the manuscript. We changed several parts of the text in order to improve the narrative, and we also tried to cover all the problems on the writing in English.
The specific comments can be found in the specific file, while the main issue with the methodology is that some sensors sound very different volumes, due to the orbit and to the very coarse spatial resolution. What is it the added value of using for example OMPS Limb profiler for example? Is the goal of the paper to study the optical properties of the plume from eruption to SP area? Or just to retrieve the optical properties at SP and then validate the observations with satellite sensors? As It is, those two objectives seem to mix up.
References in some sections are sometimes scarce, while in other sections are redundant.
Response: We included some references in order to improve the quality of the manuscript.
Point 1: On p. 4, line 153: ‘AERONET products from Level 2 version data’.
Response 1: We used version 3 - level 2 data.
Point 2: On p. 4, line 158-159: ‘The provided spatial resolution are 250 by 250 m...’.
Response 2: We used the spatial resolution values in meters just like in the reference cited.
Levy, R.C.; Remer, L.A.; Kleidman, R.G.; Mattoo, S.; Ichoku, C.; Kahn, R.; Eck, T.F. Global evaluation of the Collection 5 MODIS dark-target aerosol products over land. Atmospheric Chemistry and Physics 2010, 10, 10399–10420. doi:10.5194/acp-10-10399-2010.
Remer, L.A.; Kaufman, Y.J.; Tanré, D.; Mattoo, S.; Chu, D.A.; Martins, J.V.; Li, R.R.; Ichoku, C.; Levy, R.C.; Kleidman, R.G.; Eck, T.F.; Vermote, E.; Holben, B.N. The MODIS Aerosol Algorithm, Products, and Validation. Journal of the Atmospheric Sciences 2005, 62, 947–973, [https://doi.org/10.1175/JAS3385.1]. doi:10.1175/JAS3385.1.
Point 3: On p. 7, line 256-257: “On April 27th a volcanic plume was again identified by CALIPSO about 700 km from SPU LALINET station”. It is very thiny...are you sure it is not a cirrus? I would focus on SP area.
Response 3: According to the CALIPSO Level 2 version 4 aerosol data and Vertical Feature Mask data this layer is classified as stratospheric layer and classified as volcanic ash and sulfate type. We rewritten all the discussion of CALIPSO data on this section.
In addition, would be impossible to focus only on SP area since this is the closest overpass over SP on this day.
Point 4: On p. 13, equation 5: About the transmittance method and the equation 5, this method provides a constant value for extinction.
Response 4: Yes, the transmittance method provides a constant value for extinction for this particular aerosol layer, not for the whole aerosol extinction profile.
Point 5: On p. 13, equation 6: how the methodology is sensitive to the first lidar ratio assumed with KSF?.
Response 5: We didn’t check the sensitive to the first lidar ratio assumed with KSF since we are focus only on the aerosol layer of the volcanic plume. We just calculated the backscatter integrated value obtained by the integration of the backscatter signal between the range altitude of 18 and 19.3 km. The backscatter profile was retrieved applying the KFS method tuning the lidar ratio to the AOD value retrieved from AERONET sunphotometer.
Point 6: On p. 14, line 400: “we retrieved a lidar ratio value of 76 ± 27 sr using the equations 6 and 7”. How this value agrees with 25 sr found by CALIPSO?
Response 6: We rewritten all the CALIPSO results and discussion.

Reviewer 2 Report
The authors are describing their approach to observe high altitude aerosol layer from Calbuno Volcanic eruption. The manuscript holds scientific merit and for this reason I think it worth being published in the Remote Sensing journal of MDPI. However, in order to be improved I would kindly suggest to the authors to take into considerations the following comments:
1. Title: correct the typo to "Synergetic"
2. Line 8: Consider mentioning the full name of the sensors abbreviations (e.g. CALIPSO, MODIS, OMPS, LALINET)
3. Line 12: Delete the "should"
4. Line 16: The information of the CALIPSO dataset level is too technical I think for being in the abstract. Consider changing this sentence as follows:
"... at 532 obtained by CALIPSO, revealed the height range extension of the aerosol plume ..."
5. Line 24: "achieving" -> "achievement"
6. Line 32: The section of the Introduction should be listed as 1. Update also the number of the rest sections
7. Line 34: "due to temporal and spatial variability"
8. Line 35: "and due to the lack"
9. Line 38: "hydrological"
10. Line 42: "Mt Pinatubo eruption"
11. Line 43: "In past" -> "Over the past"
12. Line 45: "VEI" -> "Provide here the definition of this acronym and not in line 52."
13. Lines 61-64: "Various issues in this sentence. Consider revising and making it shorter."
14. Line 66: "...remote sensing techniques, both from space and ground, with measurements obtained in the path..."
15. Line 67: Here and elsewhere: Provide the full name of the acronyms in when at their first time of usage in the manuscript.
16. Line 73: Delete "should"
17. Lines 75-104 have to be revised and should be moved in the Results-Discussion section. The Introduction have to be enhanced and the scientific question answered in this manuscript have to be highlighted here.
18. Line 108: "transmission" -> "transfer"
19. Line 109-111: Consider making clearer this sentence.
20. Line 111: To highlight some special points
21. Line 124: "...backscattered by nitrogen molecules are detected at 387, 408 and 530 nm..."
22. Line 126: "are filteres" -> "background suppression is achieved"
23. Lines 134-135: "thin volcanic aerosol the transmittance method [17] was applied to determine the aerosol layer optical depth and lidar ratio."
24. Line 141: "global network" is more appropriate than "international system"
25. Lines 163-166: Revise this sentence
26. Line 170: "used in this"
27. Line 173: "with a pulse"
28. Line 175-177: Consider revising this sentence
29. Line 211: Why to utilize a version that is not evaluated yet but is still under development?
30. Line 244: "detection" -> "constellation"
31. Line 246: "The information of the air mass back trajectory height and time have to be indicated in the figure and the text."
32. Line 250: Is the lidar ratio of 25 sr representative for volcanic particles? Have in mind that CALIOP is a backscatter lidar and the direct retrieval of aerosol extinction is not feasible
33. Line 267: Figure 3 is not sufficiently explained in the manuscript. Moreover, I have the feeling that this figure is not so important since all the relevant information (track and attenuated backscatter profiles) is already given by the authors in Figure 2. I suggest to the authors to either describe Figure 3 more in the manuscript or delete it.
34. Line 285: This is the main result from the figures 4 and 5. Additionally the same pattern has been revealed also in the long range transport of Eyjafjallajokull volcanic plume over Athens Greece. See figure 7 by Kokkalis et al., 2013.
35. Line 288: "applied" -> "used "
36. Figure 4 caption: Delete the first "Aeronet".
37. Line 295-297: Revise this statement in order to make it clearer
38. Line 299: "dual aerosol"
39. Lines 299 - 340. The authors are using the word "aerosols" (plural), in many cases with no meaning. Consider correcting when appropriate.
40. Figure 7 caption: "discontinuous" -> "dashed"
41. Line 320: "facts are show" -> "behavior can be found"
42. Line 323: "0.01(units?)"
43. Line 330: The authors are kindly requested to mention from where this information comes from. It would be beneficiary for the manuscript if the wind speed is also plotted in the radiosonde temperature profile presented in figure 9
44. Line 333: "0.01(units?)"
45. Line 337: "consider"
46. Line 352: Consider providing a reference for equation 1 e.g. Weitkamp, C. (2005).
47. Line 355: "the light travel velocity" -> "the speed of light"
Line 356: "and a variable….lidar system". Which one is this variable in the Eq. 1? Also the variable O(λ,z) namely the overlap function is missing from this equation, although that for higher atmospheric observations it becomes unit. Consider revising Eq. 1. including all the terms.
Line 356-357: "…is the β(λ,z)… molecules." -> "are the backscatter β(λ,z) and the extinction α(λ,z) coefficients, which refer to both the aerosol and molecular contribution."
48. Line 365: "Radiosonde data are"
49. Line 367: "then for any value of R greater than 1"
50. Line 384: "written"- > "found"
51. Line 397: "By using"
52. Line 399: "integration of the backscatter" -> "lidar"
53. From line 403 and onwards the authors are not following the MDPI referencing style, in the manuscript. Consider updating the referencing style in the text
54. Figure 9 caption: Consider annotating this also refer to this in the manuscript with a) and b) since a) is showing the data obtained from the radiosonde and b) the lidar signal.
55. Line 435: "Mona et al., 2012". Similar to this study and for the same volcanic eruption, Kokkalis et al., 2013 revealed the optical and microphysical properties along with mass concentration of the observed volcanic plume after 4 days of travel over Athens, Greece. Consider including this as a reference.
56. Line 468: "in a certainly way" -> "definitely"
References
Weitkamp, C. (2005). LIDAR: Range-Resolved Optical Remote Sensing of the Atmosphere. Weitkamp (Ed.). Springer.
Kokkalis, P., A. Papayannis, V. Amiridis, R. E. Mamouri, I. Veselovskii, A. Kolgotin, G. Tsaknakis, N. I. Kristiansen, A. Stohl, and L. Mona, Optical, microphysical, mass and geometrical properties of aged volcanic particles observed over Athens, Greece, during the Eyjafjallajökull eruption in April 2010 through synergy of Raman lidar and sunphotometer measurements, Atmospheric Chemistry and Physics, 13, 9303-9320, doi:10.5194/acp-13-9303-2013, 2013."
Author Response
We would like to express our gratitude to the reviewer. Below our replies and in the attached file .
Dear MDPI/RS editors and reviewer #2
We would like to express our gratitude for the carefully made review. Our comments and responses are given below:
The authors are describing their approach to observe high altitude aerosol layers from Calbuno Volcanic eruption. The manuscript holds scientific merit and for this reason I think it worth being published in the Remote Sensing journal of MDPI. However, in order to be improved I would kindly suggest to the authors to take into considerations the following comments:
Point 1: Title: correct the typo to "Synergetic"
Response 1: Done as requested
Point 2: Line 8: Consider mentioning the full name of the sensors abbreviations (e.g. CALIPSO, MODIS, OMPS, LALINET)
Response 2: Done as requested
Point 3: Line 12: Delete the "should"
Response 3: Done as requested
Point 4: Line 16: The information of the CALIPSO dataset level is too technical I think for being in the abstract. Consider changing this sentence as follows:
"... at 532 obtained by CALIPSO, revealed the height range extension of the aerosol plume ..."
Response 4: Done as requested
Point 5: Line 24: "achieving" -> "achievement"
Response 5: Done as requested
Point 6: Line 32: The section of the Introduction should be listed as 1. Update also the number of the rest sections
Response 6: Done as requested
Point 7: Line 34: "due to temporal and spatial variability"
Response 7: Done as requested
Point 8: Line 35: "and due to the lack"
Response 8: Done as requested
Point 9: Line 38: "hydrological"
Response 9: Done as requested
Point 10: Line 42: "Mt Pinatubo eruption"
Response 10: Done as requested
Point 11: Line 43: "In past" -> "Over the past"
Response 11: Done as requested
Point 12: Line 45: "VEI" -> "Provide here the definition of this acronym and not in line 52."
Response 12: Done as requested
Point 13: Lines 61-64: "Various issues in this sentence. Consider revising and making it shorter."
Response 13: We changed the sentece of lines 61 to 64:
“On April 22nd 2015, the Calbuco volcano in Chile (Lat:-41.3299, Long:-72.6184) erupted after 43 years of innactivity followed by a great amount of aerosol injection into the atmosphere. The pyroclastic material dispersed into the atmosphere posed at first a threat to aviation traffic, air quality which prompted an alert in a large area, from its location to Patagonian and Pampean regions, reaching the Atlantic and Pacific Oceans and neighbouring countries, Argentina, Brazil, Paraguay and Uruguay, covered by the westerly winds at these latitudes. ”
to
“On April 22nd 2015, after 43 years of innactivity, the Calbuco volcano in Chile (Lat:-41.3299, Long:-72.6184) injected a large amount of volcanic ash aerosols into the atmosphere from a VEI 5 eruption. The pyroclastic material dispersed into the atmosphere posed at first a threat to aviation traffic and air quality, which prompted an alert in a large area, from the volcano location to Patagonian and Pampean regions, at Argentina, Chile and Paraguay. Due the general air masses circulation, the volcanic aerosol plumes traveled northeastward reaching the neighbouring countries, Uruguay and Brazil. After several days, the volcanic ash from Calbuco crossed the Atlantic Ocean, reached the South region of the African continent ans was detected at Reunion Island between 18 to 21 km (Bègue et al., 2017) ”
Bègue, N.; Vignelles, D.; Berthet, G.; Portafaix, T.; Payen, G.; Jégou, F.; Benchérif, H.; Jumelet, J.; Vernier, J.P.; Lurton, T.; Renard, J.B.; Clarisse, L.; Duverger, V.; Posny, F.; Metzger, J.M.; Godin-Beekmann, S. Long-range transport of stratospheric aerosols in the Southern Hemisphere following the 2015 Calbuco eruption. Atmospheric Chemistry and Physics 2017, 17, 15019–15036. doi:10.5194/acp-17-15019-2017.
Point 14: Line 66: "...remote sensing techniques, both from space and ground, with measurements obtained in the path..."
Response 14: Done as requested.
Point 15: Line 67: Here and elsewhere: Provide the full name of the acronyms in when at their first time of usage in the manuscript.
Response 15: Done as requested in the abstract section for CALIPSO satellite, MODIS system, OMPS system, Suomi NPP satellite, AERONET sunphotometer, and LALINET.
Point 16: Line 73: Delete "should"
Response 16: Done as requested.
Point 17: Lines 75-104 have to be revised and should be moved in the Results-Discussion section. The Introduction have to be enhanced and the scientific question answered in this manuscript have to be highlighted here.
Response 17: The lines 75-104 were revised and was moved to the results section. The introductions was re-written as:
“The presence of volcanic aerosol layers could be identified promptly at the proximities of Calbuco and hereafter by sensing techniques, both from space and ground, with measurements obtained in the path of the dispersed aerosols. CALIPSO, MODIS and OMPS onboard of Suomi NPP satellite were the space plataforms used to track the injected layers and lidar systems from LALINET network in South America allowed us to get 4-D distribution of Calbuco ashes after its occurrence (22 - 30 April). Most of lidar stations had collocated AERONET sunphotometers to help in the optical characterization and not all LALINET stations were able to observe this event given the air circulation pattern dominating this part of the globe and their distance of the atmospheric injection local. Here we present the volcanic layer transported over São Paulo area where a station is located showing the presence of erupted material around 19 km. The detection of volcanic plume in April 2015 at SPU LALINET station was the first event of this kind for this station and with the aid of remote sensing by means of remote sensing plataform data, namely: CALIPSO, MODIS, OMPS LP, the AERONET sunphotometer and SPU lidar system, we were able to estimate the sole plume AOD, its optical properties and retrieve the plume transmission, AOD, extinction and backscatter profiles, and Lidar ratio values in order to classify the volcanic aerosol. Also this observation will help to understand the mass transfer between the troposphere and lower stratosphere and models related to that [Appenzeller et al., 1996]. This observation should be an initial effort to create a lidar-based observation database like those available in many sites around the globe [Zuev et al., 2017, Khaykin et al., 2017].
In this paper we present the tracking of the volcanic plume right after its release into the atmosphere up to about five days later when it is detected over the SPU lidar station in São Paulo. As one will realize the plume at some point bisected and one portion travelled eastwards towards the South African region and the Indic Ocean [Bègue et al., 2017, Shikwambana and Sivakumar, 2018 ]. The other limb travelled to the northeast and reached the São Paulo region. This observation is important to distinguish between the two portions and despite the good amount of reports to the one travelling eastwards, the detection of the volcanic of the plume proved to be of equal interest in its characterization. It is also worth mentioning that residence time of the released material about 0.4 Tg had a climate drive on the Ozone hole in the southern polar region [Stone et al., 2017].”
Bègue, N.; Vignelles, D.; Berthet, G.; Portafaix, T.; Payen, G.; Jégou, F.; Benchérif, H.; Jumelet, J.; Vernier, J.P.; Lurton, T.; Renard, J.B.; Clarisse, L.; Duverger, V.; Posny, F.; Metzger, J.M.; Godin-Beekmann, S. Long-range transport of stratospheric aerosols in the Southern Hemisphere following the 2015 Calbuco eruption. Atmospheric Chemistry and Physics 2017, 17, 15019–15036. doi:10.5194/acp-17-15019-2017.
Appenzeller, C.; Holton, J.R.; Rosenlof, K.H. Seasonal variation of mass transport across the tropopause. Journal of Geophysical Research: Atmospheres 1996, 101, 15071–15078. doi:10.1029/96JD00821.
Zuev, V.V.; Burlakov, V.D.; Nevzorov, A.V.; Pravdin, V.L.; Savelieva, E.S.; Gerasimov, V.V. 30-year lidar observations of the stratospheric aerosol layer state over Tomsk (Western Siberia, Russia). Atmospheric Chemistry and Physics 2017, 17, 3067–3081. doi:10.5194/acp-17-3067-2017.
Khaykin, S.M.; Godin-Beekmann, S.; Keckhut, P.; Hauchecorne, A.; Jumelet, J.; Vernier, J.P.; Bourassa, A.; Degenstein, D.A.; Rieger, L.A.; Bingen, C.; Vanhellemont, F.; Robert, C.; DeLand, M.; Bhartia, P.K. Variability and evolution of the midlatitude stratospheric aerosol budget from 22 years of ground-based lidar and satellite observations. Atmospheric Chemistry and Physics 2017, 17, 1829–1845. doi:10.5194/acp-17-1829-2017.
Shikwambana, L.; Sivakumar, V. Long-range transport of volcanic aerosols over South Africa: a case study of the Calbuco volcanic eruption in Chile during April 2015. South African Geographical Journal 2018, 100, 349–363. doi:10.1080/03736245.2018.1498383.
Stone, K.A.; Solomon, S.; Kinnison, D.E.; Pitts, M.C.; Poole, L.R.; Mills, M.J.; Schmidt, A.;
Neely III, R.R.; Ivy, D.; Schwartz, M.J.; Vernier, J.P.; Johnson, B.J.; Tully, M.B.; Klekociuk, A.R.; König-Langlo, G.; Hagiya, S. Observing the Impact of Calbuco Volcanic Aerosols on South Polar Ozone Depletion in 2015. Journal of Geophysical Research: Atmospheres 2017, 122, 11,862–11,879, doi:10.1002/2017JD026987.
Point 18: Line 108: "transmission" -> "transfer"
Response 18: Done as requested. We changed the whole sentence
Point 19: Line 109-111: Consider making clearer this sentence.
Response 19: We changed the whole sentence to:
“The instruments on board each satellite plataform will be shown below and ground-based instruments such as lidar and collocated sunphotometer will be discussed as well. The synergetic use of these platforms help to understand the impact of the volcanic plume in terms of its optical properties.”
Point 20: Line 111: To highlight some special points
Response 20: Done as requested. We changed the whole sentence
Point 21: Line 124: "...backscattered by nitrogen molecules are detected at 387, 408 and 530 nm..."
Response 21: Done as requested.
Point 22: Line 126: "are filteres" -> "background suppression is achieved"
Response 22: Done as requested.
Changed the original sentence: “The photons elastically backscattered at the 355, 532 and 1064 nm wavelength and the photons inelastically (Raman) scattered by nitrogen molecules at are detected 387, 408 and 530 nm by photomultiplier tubes (PMTs, Hamamatsu type R9880U-110) and are filteres by means of interference filters with a FWHM of 1 nm at the elastic channels and 0.25 nm for the inelastic ones.” by
“The photons elastically backscattered at the 355, 532 and 1064 nm wavelength and the photons inelastically (Raman) scattered by nitrogen molecules are detected at 387 and 530 nm, and by water vapour molecules at 408 nm using photomultiplier tubes (PMTs, Hamamatsu type R9880U-110). All these backscattered signals are filtered and the background suppression is achieved by means of interference filters with a FWHM of 1 nm at the elastic channels and 0.25 nm for the inelastic ones.”
Point 23: Lines 134-135: "thin volcanic aerosol the transmittance method [17] was applied to determine the aerosol layer optical depth and lidar ratio."
Response 23: Done as requested.
Point 24: Line 141: "global network" is more appropriate than "international system"
Response 24: Done as requested.
Point 25: Lines 163-166: Revise this sentence
Response 25: Done as requested.
We changed the original sentence: “In order to derive the AOD data from MODIS/AQUA all over South America was used information from the website https://worldview.earthdata.nasa.gov/. It was selected the Deep Blue Aerosol Optical Depth layers that is useful for studying aerosol optical depth over land surfaces. These layers were created from the Deep Blue algorithm. The MODIS Deep Blue Aerosol Optical Depth (Land) layer is available from both the Terra and Aqua satellites for daytime overpasses. The sensor/algorithm resolution is 10 km at nadir, imagery resolution is 2 km at nadir, and the temporal resolution is daily. In this study 550 nm AOD product from aerosol Level 2 were used. ” by
“In order to retrieve aerosol optical properties from atmosphere, MODIS uses two different and independent algorithms, the deep blue is responsible to aerosol products for land retrieval only, and the Dark target is a separate algorithm for aerosol products over land and ocean [24,25]. To derive the AOD data from MODIS/AQUA all over South America it was selected the Deep Blue Aerosol Optical Depth layers from the website https://worldview.earthdata.nasa.gov/. The sensor/algorithm resolution used was 10 km at nadir, imagery resolution of 2 km at nadir, and daily temporal resolution. It was retrieved the AOD product at 550 nm from aerosol Level 2.”
Point 2:6 Line 170: "used in this"
Response 26: Done as requested.
Point 27: Line 173: "with a pulse"
Response 27: Done as requested.
Point 28: Line 175-177: Consider revising this sentence
Response 28: Done as requested.
We changed the original sentence: “The CALIOP data products are assembled from the backscattered signals and divided in two categories, level 1 profiles products and level 2 profiles and layers products. [26]. The level 2 products are derived from the level 1 products and three different level 2 products are distributed according to the layer products, profile products and the vertical feature mask (VMF).” by
“The CALIOP data products are assembled from the backscattered signals and divided in two categories, level 1 profiles, and level 2, which are compounded by profile and layer products. Level 1 products are used to derive level 2, which in turn, are organized in three different types, layer and profile products and the vertical feature mask (VMF) [26].”
Point 29: Line 211: Why to utilize a version that is not evaluated yet but is still under development?
Response 29: We preferred to used Version 4 data instead of version 3, because it included several improvements concerning the aerosol subtyping and lidar ratio retrieval. As we assert in the text, the most relevant improvement in Version 4 products algorithm is the possibility to identify aerosol subtypes in the stratosphere. Even if the version 4 data are still under evaluation and validation, which is the case for São Paulo, where we are still in progress of version 4 evaluation.
Point 30: Line 244: "detection" -> "constellation"
Response 30: Done as requested.
Point 31: Line 246: "The information of the air mass back trajectory height and time have to be indicated in the figure and the text."
Response 31: Done as requested.
We included an inset figure showing the air mass back trajectories height and time in figure 1 and in the text.
Point 32: Line 250: Is the lidar ratio of 25 sr representative for volcanic particles? Have in mind that CALIOP is a backscatter lidar and the direct retrieval of aerosol extinction is not feasible
Response 32: We mistakenly used the 5km aerosol layer data from version 3, which there are no algorithm to classify stratospheric layers. Each layer at the stratosphere is assigned by lidar ratio value of 25 sr. As commented on the manuscript, 5km aerosol layer data version 4 products from CALIPSO was released in November 2016 and included several improvements concerning the aerosol sub-typing and lidar ratio retrieval. The most relevant improvement in Version 4 products algorithm is the possibility to identify aerosol subtypes in the stratosphere [kim et al., 2018]. These subtypes is associated to four aerosol types, polar stratospheric aerosol (PSA), volcanic ash, sulphate/other, and smoke.
Kim, M.H.; Omar, A.H.; Tackett, J.L.; Vaughan, M.A.; Winker, D.M.; Trepte, C.R.; Hu, Y.; Liu, Z.; Poole, L.R.; Pitts, M.C.; Kar, J.; Magill, B.E. The CALIPSO version 4 automated aerosol classification and lidar ratio selection algorithm. Atmospheric Measurement Techniques 2018, 11, 6107–6135. doi:10.5194/amt-11-6107-2018.
Point 33: Line 267: Figure 3 is not sufficiently explained in the manuscript. Moreover, I have the feeling that this figure is not so important since all the relevant information (track and attenuated backscatter profiles) is already given by the authors in Figure 2. I suggest to the authors to either describe Figure 3 more in the manuscript or delete it.
Response 33: We preferred to follow the reviewer#1 suggestion and describe better figure 3, includingthe following sentence:
”The same can be checked on figure 04, where the top panel presents the 532 nm Total attenuated backscatter profile for the CALIPSO overpass distance of 700 km from SPU station from the latitude. As one can see in the top panel of figure 04 there is a aerosol plume detected between 19 to 20 km of altitude and between the latitude coordinates of -21.02 and -27.10. The same aerosol plume can be seen in the bottom panel of figure 04, where is presented the 532 nm range corrected signal retrieved by the SPU lidar station during 12:22 UTC to 21:10 UTC. The aerosol plume is detected at 19 km of altitude throughout the entire measurement period. Both graphics show the coincidence between the volcanic plume detection by both platforms, CALIPSO satellite and SPU lidar ground-based station.”
Point 34: Line 285: This is the main result from the figures 4 and 5. Additionally the same pattern has been revealed also in the long range transport of Eyjafjallajokull volcanic plume over Athens Greece. See figure 7 by Kokkalis et al., 2013.
Response 34: We would like to thank the reviewer #1 for alerting us about the excellent results presented by kokkalis et al., 2013. We really fail to let such important results go unnoticed. We include the following sentence on the manuscript:
“These important results retrieved from AERONET sunphotometer presented on figures 3 and 4 have the same pattern revealed on the study of long range transport of Eyjafjallajökull volcanic plume over Athens Greece presented by Kokkalis et al., 2013, where low AOD values of approximately 0.1 were retrieved at 500 nm before the volcanic ashes arrive, and during the advection of the plume the AOD peaked at a value of 0.25, and after the volcanic plume event the AOD values decreased again to 0.20. In addition, Kokkalis et al., 2013 shown the domination of fine mode particles during the volcanic ash detection, where fine-mode particles inside the atmospheric column were on the order of 59.1–60.9% before and after the event, and increased to 76.8–78.0% during volcanic event detection.”
Point 35: Line 288: "applied" -> "used "
Response 35: Done as requested.
Point 36: Figure 4 caption: Delete the first "Aeronet".
Response 36: Done as requested.
Point 37: Line 295-297: Revise this statement in order to make it clearer
Response 37: The original sentence:
“Although the nearest SAOD value from OMPS LP is the lowest among all the coincident the time and space (500 km radius) measurements, the magnitude of the SAOD measured by the lidar, 0.007, is an order of magnitude lower.”
was replaced by:
“All the time and space coincident SAOD measurements are an order of magnitude higher than the SAOD measured by the lidar, 0.007.”
Point 38: Line 299: "dual aerosol"
Response 38: Done as requested.
Point 39: Lines 299 - 340. The authors are using the word "aerosols" (plural), in many cases with no meaning. Consider correcting when appropriate.
Response 39: Done as requested.
Point 40: Figure 7 caption: "discontinuous" -> "dashed"
Response 40: Corrected as requested.
Point 41: Line 320: "facts are show" -> "behavior can be found"
Response 41: Corrected as requested.
Point 42: Line 323: "0.01(units?)"
Response 42: Corrected as requested.
Point 43: Line 330: The authors are kindly requested to mention from where this information comes from. It would be beneficiary for the manuscript if the wind speed is also plotted in the radiosonde temperature profile presented in figure 9
Response 43: An error in the calculus of the zonal wind was found. The error was corrected and the results of the rough analysis did not allowed to explain the disagreement between the vertical extensions of the spike-like layer measured by the lidar and the OMPS LP. A more detailed analysis was conducted determining the mean wind magnitude and direction matching the altitudes of spike-like lidar layer.
We included changes on the text from lines 330 to 340.
“The mean zonal wind of the layer from 18.5 to 20.5 km at 12:00 UTC was -12.9 km/h, noting that the maximum speed in the layer was -21.4 km/h. The lidar profile was measured at 12:00 UTC and the OMPS LP measurements were taken approximately at 16:30 UTC with a difference of 4 and a half hours. Then roughs estimates of the zonal transport of the lidar aerosols spike-like layer, measured at 46.7 W produce 60 km when we use the mean zonal wind and 100 km when using the maximum zonal wind of the layer. Those results allows to locate the spike-like layer measured by the lidar, at around 45.5 W at 16:26 UTC, the time the OMPS LP measurements were conducted. The nearest cross section that day, at 42 W shows at 24 S a broad layer extending from right below 20 km down to around 15 km, most of it not present in the lidar spike/like layer. Although the aerosol extinction of the layer shown in the OMPS LP cross section match the magnitude of aerosol extinction in the lidar measured layer, they do not agree in their vertical extension.
Then a more precise and detailed transport analysis was conducted using the W-E and S-N components from the two sounding levels located between the base and top of the spike-like lidar measured layer. The magnitude of the mean wind of the layer was 19.4 km/h and its direction 167.1 degrees. Then at 16:26 UTC the spike-like layer will be located around a degree in latitude south of the station, approximately 25 S. In the 42 W OMPS LP cross section that day, at 25 S the layer is narrow, with its top below 20 km and its base above 18 km. The former analysis helps to explain the apparent disagreement between the lidar and OMPS LP measurements, regarding the vertical extension of the layer.”
We also included the plot of wind speed together with the radiosonde temperature.
Point 44: Line 333: "0.01(units?)"
Response 44: Corrected as requested.
Point 45: Line 337: "consider"
Response 45: Corrected as requested.
Point 46: Line 352: Consider providing a reference for equation 1 e.g. Weitkamp, C. (2005).
Response 46: Done as requested.
Point 47: Line 355: "the light travel velocity" -> "the speed of light"
Response 47: Done as requested.
Point 47b: Line 356: "and a variable….lidar system". Which one is this variable in the Eq. 1? Also the variable O(λ,z) namely the overlap function is missing from this equation, although that for higher atmospheric observations it becomes unit. Consider revising Eq. 1. including all the terms.
Response 47b: Done as requested.
We included the variable η related to the efficiency of the lidar system in the equation 1, and also the overlap function O(λ,z).
Point 47c: Line 356-357: "…is the β(λ,z)… molecules." -> "are the backscatter β(λ,z) and the extinction α(λ,z) coefficients, which refer to both the aerosol and molecular contribution."
Response 47c: Corrected as requested.
Point 48: Line 365: "Radiosonde data are"
Response 48: Corrected as requested.
Point 49: Line 367: "then for any value of R greater than 1"
Response 49: Corrected as requested.
Point 50: Line 384: "written"- > "found"
Response 50: Corrected as requested.
Point 51: Line 397: "By using"
Response 51: Corrected as requested.
Point 52: Line 399: "integration of the backscatter" -> "lidar"
Response 52: Corrected as requested.
Point 53: From line 403 and onwards the authors are not following the MDPI referencing style, in the manuscript. Consider updating the referencing style in the text
Response 53: Corrected as requested.
Point 54: Figure 9 caption: Consider annotating this also refer to this in the manuscript with a) and b) since a) is showing the data obtained from the radiosonde and b) the lidar signal.
Response 54: Corrected as requested.
Point 55: Line 435: "Mona et al., 2012". Similar to this study and for the same volcanic eruption, Kokkalis et al., 2013 revealed the optical and microphysical properties along with mass concentration of the observed volcanic plume after 4 days of travel over Athens, Greece. Consider including this as a reference.
Response 55: As suggested by the reviewer #2 we included the following sentence in the manuscript:
“Similar to this study and for the same volcanic eruption, Kokkalis et al., 2013 revealed the optical and microphysical properties along with mass concentration of the observed volcanic plume after 4 days of travel over Athens, Greece. In this study, aerosol plumes detected between 1 to 2.3 km presented lidar ratio values of 80 ± 1 sr and 77 ± 3 sr for 355 and 532 nm, respectively. Volcanic plumes between 2.5 to 3 km presented lidar ratio values of 78 ± 1 and 76 ± 5 sr for 355 and 532 nm, respectively; and high plumes detected at 5 to 6 km presented lidar ratio values of 60 ± 2 and 44 ± 8 sr for 355 and 532 nm, respectively.”
Point 56: Line 468: "in a certainly way" -> "definitely"
Response 56: Done as requested.
References
Point 57: Weitkamp, C. (2005). LIDAR: Range-Resolved Optical Remote Sensing of the Atmosphere. Weitkamp (Ed.). Springer.
Response 57: we included the reference as requested.
Point 58: Kokkalis, P., A. Papayannis, V. Amiridis, R. E. Mamouri, I. Veselovskii, A. Kolgotin, G. Tsaknakis, N. I. Kristiansen, A. Stohl, and L. Mona, Optical, microphysical, mass and geometrical properties of aged volcanic particles observed over Athens, Greece, during the Eyjafjallajökull eruption in April 2010 through synergy of Raman lidar and sunphotometer measurements, Atmospheric Chemistry and Physics, 13, 9303-9320, doi:10.5194/acp-13-9303-2013, 2013."
Response 58: we included the reference as requested.
The author and co-authors would like to thank the reviewer#2 for all comments, suggestions and references in order to improve the manuscript. In included both references suggested.

Reviewer 3 Report
See an attached pdf file.

Author Response
We would like to express our gratitude for the carefully made review. Our comments and
responses are given on the attached file.

Round 2
Reviewer 1 Report
I am happy that the reviewers addressed all the issues I raised during the first revision and now the paper is ready for publication after some minor changes.
In literature the word forcing is often misused. Actually forcing is referring to pre-industrial era, while in this study "effect" is more appropriate. Please correct it through the whole manuscript.
Line 113 pag 2: please read "in the next sections" instead of "below"
Line 120 pag 3:please read "Lidars are instruments that use a laser beam as source(25). For the lidar system at SPU station, a commercial Nd:YAG laser operating at 1064, 532 and 355 nm is used.
Line 141 pag 4. I encourage to add the following reference to make the manuscript more detailed. Please read:"Each different lidar technique has a different impact on the retrieval (ref: https://doi.org/10.5194/amt-11-1639-2018)
Author Response
Dear Sir or Madam,
Thank you once more for your careful review.
We have put minor revision in the text as asked.
Our best regards.

Reviewer 2 Report
The authors addressed sufficiently my concerns and followed my comments and suggestions throughout the manuscript. Therefore, in my opinion the revised version of the manuscript can be published the Remote Sensing journal of MDPI, in its present form.
Author Response

(The authors gave the same response as above.)
